# RePo: Resilient Model-Based Reinforcement Learning by Regularizing Posterior Predictability

**Chuning Zhu**
University of Washington
Seattle, WA 98105
zchuning@cs.washington.edu

**Max Simchowitz**
Massachusetts Institute of Technology
Boston, MA 02139
msimchow@mit.edu

**Siri Gadipudi**
University of Washington
Seattle, WA 98105
sg06@uw.edu

**Abhishek Gupta**
University of Washington
Seattle, WA 98105
abhgupta@cs.washington.edu

## Abstract

Visual model-based RL methods typically encode image observations into low-dimensional representations in a manner that does not eliminate redundant information. This leaves them susceptible to *spurious variations* – changes in task-irrelevant components such as background distractors or lighting conditions. In this paper, we propose a visual model-based RL method that learns a latent representation resilient to such spurious variations. Our training objective encourages the representation to be maximally predictive of dynamics and reward, while constraining the information flow from the observation to the latent representation. We demonstrate that this objective significantly bolsters the resilience of visual model-based RL methods to visual distractors, allowing them to operate in dynamic environments. We then show that while the learned encoder is resilient to spurious variations, it is not invariant under significant distribution shift. To address this, we propose a simple reward-free alignment procedure that enables test time adaptation of the encoder. This allows for quick adaptation to widely differing environments without having to relearn the dynamics and policy. Our effort is a step towards making model-based RL a practical and useful tool for dynamic, diverse domains. We show its effectiveness in simulation benchmarks with significant spurious variations as well as a real-world egocentric navigation task with noisy TVs in the background. Videos and code: https://zchuning.github.io/repo-website/.

## 1 Introduction

Consider the difference between training a single robot arm against a plain background with reinforcement learning (RL), and learning to operate the same arm amidst of plentiful dynamic distractors - uncontrollable elements such as changing lighting and disturbances in the scene. The latter must contend with *spurious variations* - differences in environments which are irrelevant for the task but potentially confusing for a vision-based RL agent - resilience to which is indispensable for truly versatile embodied agents deployed in real world settings.

Standard end-to-end techniques for visual RL struggle in the presence of spurious variations [64, 48], in part because they fail to discard task-irrelevant elements. To improve generalization [38, 59], self-supervised representation learning methods [23, 39, 55, 54, 17, 31] pre-train visual encoders that compress visual observations. These methods aim for lossless compression of how image observations evolve in time (e.g. by minimizing reconstruction error). Unaware of the demands of downstream

37th Conference on Neural Information Processing Systems (NeurIPS 2023).

tasks, these methods also cannot determine which elements of an environment can be discarded. As such, they often struggle in dynamic and diverse scenes [64, 48, 17] - ones where significant portions of the observations are both unpredictable and irrelevant - despite being remarkably successful in static domains.

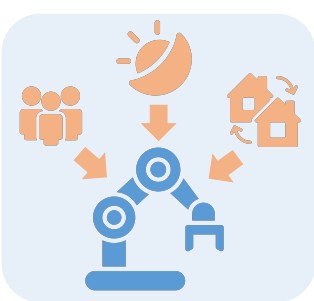

This paper proposes Resilient Model-Based RL by **Re**gularizing **Po**steior Predictability (RePo) – an algorithm for learning lossy latent representations resilient to spurious variations. A representation is satisfactory if it (a) predicts its own dynamics and (b) accurately predicts the reward. To satisfy these criteria, RePo jointly learns (i) a visual encoder mapping high-dimensional observations to intermediate image "encodings" (ii) a latent encoder which compresses histories of intermediate image encodings into compressed *latent representations* (iii) a dynamics model in the latent representation space, and (iv) a reward predictor to most accurately predict current and future rewards. What distinguishes us from past work [63, 12, 17] is a new desideratum of *predictability:* that, conditioned on past latents and actions, future latent dynamics should look *as deterministic as possible*. This is because an agent should try to maximize its control over task-relevant parts of the state, whilst neglecting aspects of the environment that it cannot influence [20, 60]. RePo optimizes a novel loss which encourages *predictability*, thereby discarding a broad range of spurious variations in aspects of the environment

Figure 1: Reinforcement learning in environments with spurious variations - including dynamic elements like humans, changes in lighting and training across a range of visual appearances.

which are out of the agents control (e.g. changes in background, lighting, or visual traffic in the background). At the same time, by penalizing reward prediction error, we capture the *task-relevant* aspects of the dynamics necessary for learning performant policies.

RePo implements a deceptively simple modification to recurrent state-space models for model-based RL [17, 59, 46]. We maximize mutual information (MI) between the current representation and *all* future rewards, while minimizing the mutual information between the representation and observation. Instead of minimizing image reconstruction error, we optimize a variational lower bound on the MI-objective which tractably enforces that the learned observation encoder, latent dynamics and reward predictors are highly informative of reward, while ensuring latents are as *predictable* as possible (in the sense described above). We demonstrate that the representations, and the policies built thereupon, learned through RePo succeed in environments with significant amounts of dynamic and uncontrollable distractors, as well as across domains with significant amounts of variability and complexity. Through ablations, we also validate the necessity of our careful algorithm design and optimization decisions.

While these learned representations enable more effective reinforcement learning in dynamic, complex environments, the visual encoders (point (i) above) mapping from observations into intermediate encodings suffer from distribution shift in new environments with novel visual features (e.g. a new background not seen at train time.) We propose a simple test-time adaptation scheme which uses (mostly) unlabeled test-time data to adapt the *visual encoders* only, whilst keeping all other aspects of the RePo model fixed. Because RePo ensures resilience of the compressed latent representation at training time, modifying only the test-time visual encoders to match training time representations allows representations to recover optimal performance with only minor amounts of adaptation.

Concretely, the key contributions of this work are: **(1)** We propose a simple representation learning algorithm RePo for learning representations that are informative of rewards, while being as predictable as possible. This allows model-based RL to scale to dynamic, cluttered environments, avoiding reconstruction. **(2)** We show that while the learned encoders may be susceptible to distribution shift, they are amenable to a simple test-time adaptation scheme that can allow for quick adaptation in new environments. **(3)** We demonstrate the efficacy of RePo on a number of simulation and real-world domains with dynamic and diverse environments.

## 2 Related Work

Our work is related to a number of techniques for visual model-based reinforcement learning, but differs in crucial elements that allow it to scale to dynamic environments with spurious variations.

**Model-Based RL.** Though model-based RL began with low-dimensional, compact state spaces [26, 37, 27, 57], advances in visual model-based reinforcement learning [17, 19, 18, 44, 42, 21] learn latent representations and dynamics models from high dimensional visual feedback (typically via recurrent state-space models). Perhaps most relevant to RePo is DREAMER [17]. Section 4 explains the salient differences between DREAMER and RePo; notably, we eschew a reconstruction loss in pursuit of resilience to spurious variations. A closely related work is TD-MPC [22], which learns a task-oriented latent representation by predicting the value function. However, its representation may not discard irrelevant information and necessarily contains information about the policy.

**Representation Learning for Control.** There is a plethora of techniques for pretraining visual representations using unsupervised learning objectives [38, 34, 30, 32, 41, 49, 47, 13]. While these can be effective on certain domains, they do not take downstream tasks into account. Task-relevant representation learning for RL uses the reward function to guide representation learning, typically in pursuit of *value-equivalence* (e.g. via bisimulation) [63, 8, 62, 12, 50, 22]. However, these approaches do little to explicitly counteract spurious variations. Our work aligns with a line of work that disentangles task-relevant and task-irrelevant components of the MDP. [7, 6] obtain provable guarantees for representation learning with exogeneous distractors - parts of the state space whose dynamics is independent of the agent's actions. [56] introduces a more granular decomposition of the MDP across the task relevance and controllability axes. Our work, in contrast, does not impose a specific form on the spurious variations.

**Domain Adaptation.** Unsupervised domain adaptation adapts representations across visually different source and target domains [66, 58, 45, 11, 25]. These techniques predominantly adapt visual encoders by minimizing a distribution measure across source and training distributions, such as MMD [5, 33, 53, 24], KL divergence [67, 35] or Jensen-Shannon divergence [11, 52]. In [61], distribution matching was extended to sequential decision making. While domain adaptation settings typically assume that the source and target share an underlying marginal or joint distribution in a latent space, this assumption does not hold in online RL because the data is being collected incrementally through exploration, and hence the marginals may not match. Hence, our test-time adaptation technique, as outlined in Section 4.1, introduces a novel support matching objective that enforces the test distribution to be in support of the train distribution, without trying to make the distributions identical.

## 3 Preliminaries

**MDPs.** A (discounted) MDP $\mathcal{M} = (\mathcal{S}, \mathcal{A}, \gamma, P, P_0, r)$ consists of a state-space $\mathcal{S}$, action space $\mathcal{A}$, discount factor, $\gamma \in (0, 1)$, transition and $P(\cdot, \cdot) : \mathcal{S} \times \mathcal{A} \to \triangle(\mathcal{S})$, initial state distribution $P_0 \in \triangle(\mathcal{S})$, and reward function $r(\cdot, \cdot) : \mathcal{S} \times \mathcal{A} \to [0, 1]$ (assumed deterministic for simplicity). A policy $\pi : \mathcal{S} \to \triangle(\mathcal{A})$ is a mapping from states to distributions over actions. We let $\mathbb{E}^{\pi}_{\mathcal{M}}$ denote expectations under $s_0 \sim P_0$, $a_t \sim \pi(s_t)$, and $s_{t+1} \sim P(s_t, a_t)$; the value is $V^{\pi}_{\mathcal{M}}(s) := \mathbb{E}^{\pi}_{\mathcal{M}} \left[ \sum_{t=0}^{\infty} \gamma^h r(s_t, a_t) \mid s_0 = s \right]$, and $V^{\pi}_{\mathcal{M}} = \mathbb{E}_{s_0 \sim P_0}[V^{\pi}_{\mathcal{M}}(s_0)]$. The goal is to learn a policy $\pi$ that maximizes the sum of expected returns $\mathbb{E}^{\pi}_{\mathcal{M}} \left[ \sum_{t=0}^{\infty} \gamma^h r(s_t, a_t) \mid s_0 = s \right]$, as in most RL problems, but we do so based on a belief state as explained below.

**Visual RL and Representations.** For our purposes, we take states $s_t$ to be visual observations $s_t \equiv o_t \in \mathcal{O}$; for simplicity, we avoid explicitly describing a POMDP formulation - this can be subsumed either by introducing a belief-state [68], or by assuming that images (or sequences thereof, e.g. to estimate velocities) are sufficient to determine rewards and transitions [36]. The states $o_t$ may be high-dimensional, so we learn encoders $h : \mathcal{O} \to \mathcal{X}$ to an encoding space $\mathcal{X}$. We compress these encodings $x_t$ further into latent states $z_t$, described at length in our method in Section 4.

**Spurious variation.** By *spurious variation*, we informally mean the presence of features of the states $s_t$ which are irrelevant to our task, but which do vary across trajectories. These can take the form of explicit *distractors* - either *static* objects (e.g. background wall-paper) or *dynamic* processes (e.g. video coming from a television) that do not affect the part of the state space involved in our task [7, 6]. Spurious variation can also encompass processes which are not so easy to disentangle with the state: for example, lighting conditions will affect all observations, and hence will affect the appearance of transition dynamics.

Consider the following canonical example: an MDP with state space $\mathcal{S}_1 \times \mathcal{S}_2$, where for $s = (s^{(1)}, s^{(2)}) \in \mathcal{S}_1 \times \mathcal{S}_2$, the reward $r(s, a)$ is a function $\bar{r}(s^{(1)}, a)$ only of the projection onto $\mathcal{S}^1$. Moreover, suppose that $\mathbb{P}[(s^+)^{(1)} \in \cdot \mid s, a]$, where $s^+ \sim P(s, a)$, is a distribution $\bar{P}(s^{(1)}, a)$ again only depending on $s^{(1)}$. Then, the states $s^{(2)}$ can be viewed as spuriously various. For example, if

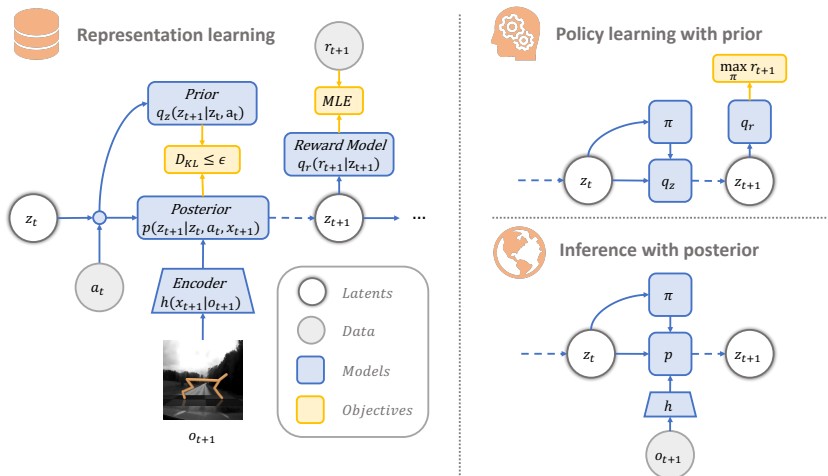

Figure 2: `RePo` learns a latent representation resilient to spurious variations by predicting the dynamics and the reward while constraining the information flow from images.

$s^{(1)}$ is a Lagrangian state and $s^{(2)}$ is a static background, then it is clear that transitions of Lagrangian state and reward do not depend on $s^{(2)}$. Our template also encompasses dynamic distractors; e.g. a television show in the background has its own dynamics, and these also do not affect reward or physical dynamics. Even varying lighting conditions can be encompassed in this framework: the shadows in a scene or brightness of the environment should not affect reward or physics, even though these visual features themselves evolve dynamically in response to actions and changes in state. That is, there are examples of spurious variation where $s^{(1)}$ (e.g. Lagrangian state) affect $s^{(2)}$ (e.g. certain visual features), but not the other way round. In all cases, "spurious" implies that states $(s_t^{(2)})_{t \geq 0}$, and their possible variations due to different environments, have no bearing on optimal actions.

## 4 `RePo`: Parsimonious Representation Learning without Reconstruction

We propose a simple technique for learning task-relevant representations that encourages parsimony by removing all information that is neither pertinent to the reward nor the dynamics. Such representations discard information about spurious variations, while retaining the information actually needed for decision making.

To describe our method formally, we introduce some notation (which is also shown in Fig 2). Let $\mathcal{O}$ be the space of image observations, $\mathcal{X}$ the space of encoded observations, where $h : \mathcal{O} \to \mathcal{X}$ represents the encoding function from images observations to encoded observations, and $\mathcal{Z}$ the space of latent representations. Note that $x_{t+1}$ is simply the instantaneous encoding of the image $o_{t+1}$ as $x_{t+1} = h(o_{t+1})$, but the latent representation $z_{t+1}$ at time step $t+1$ is an aggregation of the current encoding $x_{t+1}$ and previous latent $z_t$ and action $a_t$. Let $\mathscr{P}_{\text{post}}$ denote the space of "posteriors" on latent dynamics $z$ of the form $p(z_{t+1} \in \cdot \mid z_t, a_t, x_{t+1})$, where $z_t, z_{t+1} \in \mathcal{Z}, a_t \in \mathcal{A}, x_{t+1} \in \mathcal{X}$, and where and $z_0 \sim p_0$ has some initial distribution $p_0$. In words, the latent posterior use past latent state and action, in addition to *current encoding* to determine current latent. Control policies and learned dynamics models act on this latent representation $z_{t+1}$, and not simply the image encoding $x_{t+1}$ so as to incorporate historical information.

Let $\mathcal{D}_{\text{buf}}$ denote the distribution over experienced actions, observations and rewards from the environment $((a_{1:T}, o_{1:T}, r_{1:T}) \sim \mathcal{D}_{\text{buf}})$. For $p \in \mathscr{P}_{\text{post}}$, let $\mathbb{E}_{p,h}$ denote expectation of $(a_{1:T}, o_{1:T}, r_{1:T}) \sim \mathcal{D}_{\text{buf}}$, $x_t = h(o_t)$ and the latents $z_{t+1} \sim p(\cdot \mid z_t, a_t, x_{t+1})$ drawn from the latent posterior, with the initial latent $z_0 \sim p_0$. Our starting proposal is to optimize the latent posterior $p$ and image encoder $h$ such that information between the latent representation and future reward is maximized, while bottlenecking [1] the information between the latent and the observation:

$$\max_{p,h} \mathrm{I}_{p,h}(z_{1:T}; r_{1:T} \mid a_{1:T}) \ \text{ s.t. } \ \mathrm{I}_{p,h}(z_{1:T}; o_{1:T} \mid a_{1:T}) < \epsilon. \tag{4.1}$$

Above, $\mathrm{I}_{p,h}(z_{1:T}; r_{1:T} \mid a_{1:T})$ denotes mutual information between latents and rewards conditioned actions under the $\mathbb{E}_{p,h}$ distribution, and distribution $\mathrm{I}_{p,h}(z_{1:T}; o_{1:T} \mid a_{1:T})$ measures information

between latents and observations under $\mathbb{E}_{p,h}$ as well. Thus, (4.1) aims to preserve large mutual information with rewards whilst minimizing information stored from observations.

Optimizing mutual information is intractable in general, so we propose two variational relaxations of both objects (proven in Appendix B)

$$I_{p,h}(z_{1:T}; r_{1:T} \mid a_{1:T}) \geq \mathbb{E}_{p,h}\left[\sum_{t=1}^{T} \log q_{\mathrm{r}}(r_t \mid z_t)\right] \tag{4.2}$$

$$I_{p,h}(z_{1:T}; o_{1:T} \mid a_{1:T}) \leq \mathbb{E}_{p,h}\left[\sum_{t=0}^{T-1} D_{\mathrm{KL}}(p(\cdot \mid z_t, a_t, x_{t+1}) \parallel q_{\mathrm{z}}(\cdot \mid z_t, a_t))\right], \tag{4.3}$$

where $q_{\mathrm{r}}$ and $q_{\mathrm{z}}$ are variational families representing beliefs over rewards $r_t$ and latent representations $z_{t+1}$, respectively. We refer to $z_{t+1} \sim p(\cdot \mid z_t, a_t, x_{t+1})$ as the *latent posterior*, because it conditions on the latest encoded observation $x_{t+1} = h(o_{t+1})$. We call the variational approximation $q_{\mathrm{z}}(\cdot \mid z_t, a_t)$ the *latent prior* because it does not use the current observation $o_{t+1}$ (or it's encoding $x_{t+1}$) to determine $z_{t+1}$. Note that the right hand side of Eq. (4.3) depends on $h$ through $x_{t+1} = h(o_{t+1})$, and thus gradients of this expression incorporate gradients through $h$.

**The magic of Eq. (4.3).** The upper bound in (4.3) reveals a striking feature which is at the core of our method: that, in order to reduce extraneous information in the latents $z_t$ about observations $o_t$, it is enough to match the latent posterior $z_{t+1} \sim p(\cdot \mid z_t, a_t, x_{t+1})$ to our latent prior $q_{\mathrm{z}}(\cdot \mid z_t, a_t)$ that *does not condition on current* $x_{t+1}$. Elements that are spurious variations can be captured by $p(\cdot \mid z_t, a_t, x_{t+1})$, but not by $q_{\mathrm{z}}(\cdot \mid z_t, a_t)$, since $q_{\mathrm{z}}$ is not informed by the latest observation encoding $x_{t+1}$, and spurious variations are not predictable. To match the latent posterior and the latent prior, the latent representation must omit these spurious variations. For example, in an environment with a TV in the background, removing the TV images reduces next-step stochasticity of the environment. Thus, (4.3) encourages representations to omit television images.

**The relaxed bottleneck.** The above discussion may make it seem as if we suffer in the presence of task-relevant stochasticity. However, by replacing the terms in Eq. (4.1) with their relaxations in Eqs. (4.2) and (4.3), we only omit the stochasticity that is not useful for reward-prediction. We make these substitutions, and move to a penalty-formulation amenable to constrained optimization methods like dual-gradient descent [2]. The resulting objective we optimize to learn the latent posterior $p$, latent prior $q_{\mathrm{z}}$, reward predictor $q_{\mathrm{r}}$ and observation encoder $h$ jointly is:

$$\max_{p, q_{\mathrm{r}}, q_{\mathrm{z}}, h} \min_{\beta} \mathbb{E}_{p,h}\left[\sum_{t=1}^{T} \log q_{\mathrm{r}}(r_t \mid z_t)\right] + \beta\left(\mathbb{E}_{p,h}\left[\sum_{t=0}^{T-1} D_{\mathrm{KL}}(p(\cdot \mid z_t, a_t, x_{t+1}) \parallel q_{\mathrm{z}}(\cdot \mid z_t, a_t))\right] - \epsilon\right). \tag{4.4}$$

**Implementation details.** We parameterize $p$ and $q$ using a recurrent state-space model (RSSM) [17]. The RSSM consists of an encoder $h_\theta(x_t \mid o_t)$, a latent dynamics model $q_\theta(z_{t+1} \mid z_t, a_t)$ corresponding to the prior, a representation model $p_\theta(z_{t+1} \mid z_t, a_t, x_{t+1})$ corresponding to the posterior, and a reward predictor $q_\theta(r_t \mid z_t)$. We optimize (4.4) using dual gradient descent. In addition, we use the KL balancing technique introduced in Dreamer V2 [19] to balance the learning of the prior and the posterior. Concretely, we compute the KL divergence in Eq. (4.4) as $D_{\mathrm{KL}}(p \parallel q) = \alpha D_{\mathrm{KL}}(\lfloor p \rfloor \parallel q) + (1 - \alpha)D_{\mathrm{KL}}(p \parallel \lfloor q \rfloor)$, where $\lfloor \cdot \rfloor$ denotes the stop gradient operator and $\alpha \in [0, 1]$ is the balancing parameter. With the removal of reconstruction, the KL balancing parameters becomes especially important as shown by our ablation in Sec. 5.

**Policy learning** As is common in the literature on model-based reinforcement learning [19, 17, 18], our training procedure alternates between (1) *Representation Learning:* learning a representation $z$ by solving the optimization problem outlined in Eq. (4.4) to infer a latent posterior $p(z_{t+1} \mid z_t, a_t, x_{t+1})$, a latent prior $q_{\mathrm{z}}(z_{t+1} \mid z_t, a_t)$, an encoder $x_t = h(o_t)$ and a reward predictor $q_{\mathrm{r}}(r_t \mid z_t)$, and (2) *Policy Learning:* using the inferred representation, dynamics model and reward predictor to learn a policy $\pi_\phi(a_t \mid z_t)$ for control. With the latent representation and dynamics model, we perform actor-critic policy learning [16, 10] by rolling out trajectories in the latent space. The critic $V_\psi(z)$ is trained to predict the discounted cumulative reward given a latent state, and the actor $\pi_\phi(a \mid z)$ is trained to take the action that maximizes the critic's prediction. While policy learning is carried out entirely using the latent prior as the dynamics model, during policy execution (referred to as inference in Fig. 2), we infer the posterior distribution $p(z_{t+1} \mid z_t, a_t, x_{t+1})$ over latent representations from the current observation, and use this to condition the policy acting in the world. We refer readers to Appendix C for further details.

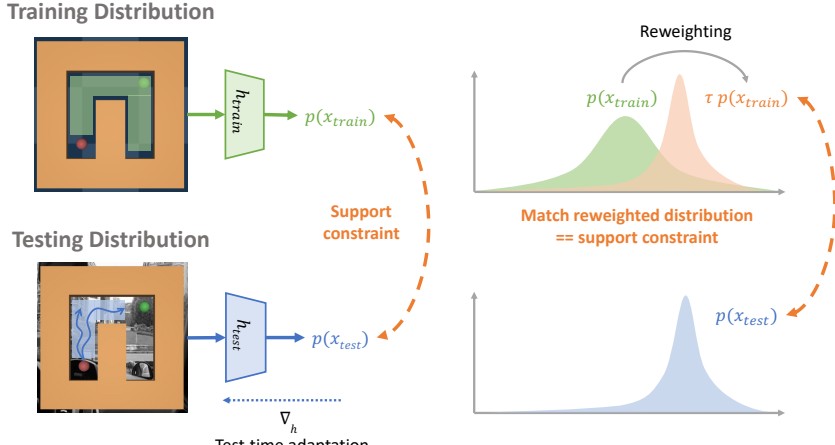

Figure 3: Depiction of test-time adaptation scheme for latent alignment via support constraints. During exploration, the marginal distributions may not match perfectly, so we match the supports of the latent features instead, using a *reweighted* distribution constraint.

**Comparison to DREAMER, DEEPMDP, and BISIMULATION.** DREAMER [17] was first derived to optimize pixel-reconstruction, leading to high-fidelity dynamics but susceptibility to spurious variations. Naively removing pixel reconstruction from dreamer, however, leads to poor performance [17]. Our objective can be interpreted as modifying DREAMER so as to maintain sufficiently accurate dynamics, but without the fragility of pixel-reconstruction. DEEPMDP [12] sets the latents $z_t$ to exactly the image encodings $x_t = h(o_t)$. It learns a dynamics $\bar{P} : \mathcal{X} \times \mathcal{A} \to \triangle(\mathcal{X})$ such that the distribution $\bar{x}_{t+1} \sim \bar{P}(h(o_t), a_t)$ is close to $x_{t+1} \sim h(o_{t+1}), o_{t+1} \sim P^\star(o_t, a_t)$, where $P^\star$ denotes a ground-truth transition dynamics; this enforces consistency of dynamics under encoding. The above distributions are viewed as conditional on *past* observation and action, and as a result, highly non-parsimonious representations such as the identity are valid under this objective. BISIMULATION [63] learns an optimal representation in the sense that a perfect bisimulation metric does not discard any relevant information about an MDP. However, there is no guarantee that it will disregard irrelevant information. Indeed, the identity mapping induces a trivial bisimulation metric. Hence, BISIMULATION compress only by reducing the dimensionality of the latent space. In contrast, we further compress the encodings $x_t$ into latents $z_t$ so as to enforce the latent prior $q_z(\cdot \mid a_t, z_t)$ is close to the latest observation-dependent posterior distribution $p(\cdot \mid z_t, a_t, x_{t+1})$. As mentioned in Eq. (4.3), this ensures information compression and invalidates degenerate representations such as the identity mapping.

### 4.1 Transferring Invariant Latent Representations via Test-Time Adaptation

While resilient to spurious variations seen during training, our learned latents $z_t$ - and hence the policies which depend on them - may not generalize to new environment which exhibit systematic distribution shift, e.g. lighting changes or background changes. The main source of degradation is that encoder $h : \mathcal{O} \to \mathcal{X}$ may observe images that it has not seen at train time; thus the latent, which depend on observations through $x_t = h(o_t)$, may behave erratically, even when system dynamics remain unchanged.

Relying on the resilience of our posteriors $p$ over latents $z_t$ introduced by RePo, we propose a test-time adaption strategy to only adjust the encoder $h$ to the new environment, whilst leaving $p$ fixed. A natural approach is to apply unsupervised domain adaptation methods [66, 58] to adapt the visual encoder $h$ to $h_{\text{test}}$. These domain adaptation techniques typically operate in supervised learning settings, and impose distributional constraints between source and target domains [61, 25], where the distributions of training and test data are stationary and assumed to be the same in *some* feature space. A distribution matching constraint would be:

$$\min_{h_{\text{test}}(\cdot)} \mathrm{D}(\mathcal{P}_{\text{train}} \parallel \mathcal{P}_{\text{test}}) \text{ s.t. } \mathcal{P}_{\text{test}} = h_{\text{test}} \circ \mathcal{D}_{\text{test}}, \mathcal{P}_{\text{train}} = h \circ \mathcal{D}_{\text{train}}. \tag{4.5}$$

In Eq. (4.5), we consider matching the distributions over encodings $x$ of observations $o$. Specifically, we assume $\mathcal{D}_{\text{train}}$ and $\mathcal{D}_{\text{test}}$ denote training and test-buffer distributions over observations $o$, $\mathcal{P}_{\text{train}} = h_{\text{train}} \circ \mathcal{D}_{\text{train}}$ denotes the distribution of $x = h_{\text{train}}(o)$ where $o \sim \mathcal{D}_{\text{train}}$ is encoded by the train-time

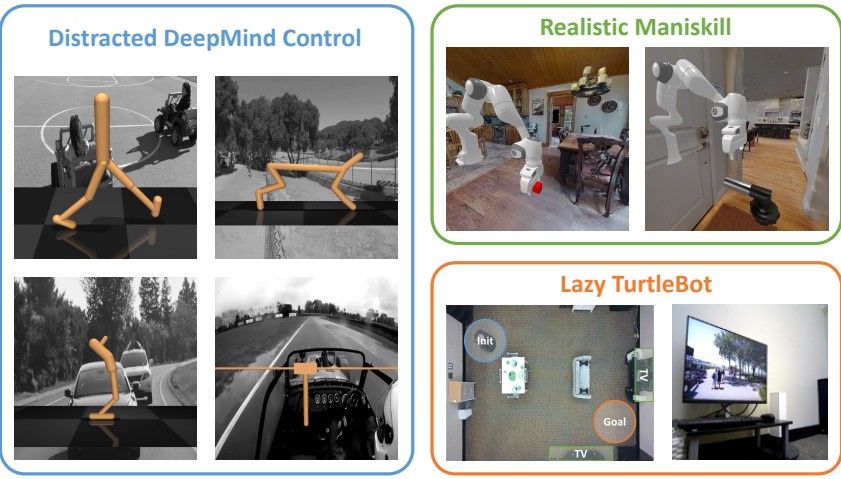

Figure 4: Depiction of the environments being used for evaluation. **(Left):** the Distracted DeepMind Control suite [64], **(Top Right)**: Maniskill2 [15] environments with realistic backgrounds from Matterport [3]. **(Bottom Right)**: TurtleBot environment with two TVs playing random videos in the background.

encoder $h_{\text{train}}$, and $\mathcal{P}_{\text{test}} = h_{\text{test}} \circ \mathcal{D}_{\text{test}}$ denotes encodings under a test-time encoder $h_{\text{test}}(\cdot)$ over which we optimize. Here, $\mathrm{D}(\cdot, \cdot)$ denotes an $f$-divergence, such as the $\chi^2$-divergence.

**Support Constraint.** (4.5) fails to capture that the encoded distributions at train and test time *differ* at the start of our adaption phase: suboptimal encoder performance at the start of the adaptation phase causes the policy to visit sub-optimal regions of state space not seen at train time. Thus, it may be impossible to match the distribution as in standard unsupervised domain adaptation. We therefore propose to replace (4.5) with a *support constraint*, enforcing that the distribution of $h_{\text{test}} \circ \mathcal{D}_{\text{test}}$ is contained in the *support* of $h_{\text{train}} \circ \mathcal{D}_{\text{train}}$. We consider the following idealized objective:

$$\min_{\tau(\cdot) \geq 0, h_{\text{test}}(\cdot)} \mathrm{D}(\tau \cdot \mathcal{P}_{\text{train}} \,\|\, \mathcal{P}_{\text{test}}) \ \text{s.t.} \ \mathbb{E}_{x \sim \mathcal{P}_{\text{train}}}[\tau(x)] = 1. \quad (4.6)$$

Here, by $\tau \cdot \mathcal{P}_{\text{train}}$, we mean the re-weighted density of $\mathcal{P}_{\text{train}} = h_{\text{train}} \circ \mathcal{D}_{\text{train}}$ by a function $\tau(x)$. The constraints $\mathbb{E}_{\mathcal{P}_{\text{train}}}[\tau(x)] = 1$ and $\tau(\cdot) \geq 0$ ensures this reweighted distribution is also a valid probability distribution. The reweighting operation $\tau \cdot \mathcal{P}_{\text{train}}$ seems intractable at first, but we show that if we take $\mathrm{D}(\cdot, \cdot) = \chi^2(\cdot, \cdot)$ to be the $\chi^2$ divergence, then Eq. (4.6) admits the following tractable Lagrangian formulation (we refer readers to [65] and Appendix B for a thorough derivation)

$$\min_{\tau(\cdot) \,\geq\, 0, h_{\text{test}}(\cdot)} \max_{f(\cdot), \lambda} \mathbb{E}_{\mathcal{P}_{\text{train}}}[\tau(x) \cdot f(x)] - \mathbb{E}_{\mathcal{P}_{\text{test}}}\left[f(x) + \frac{1}{4}f(x)^2\right] + \lambda(\mathbb{E}_{\mathcal{P}_{\text{train}}}[\tau(x)] - 1), \quad (4.7)$$

where above, $\lambda \in \mathbb{R}$, $f : \mathcal{X} \to \mathbb{R}$, and the objective depends on $h_{\text{test}}$ through the definition $\mathcal{P}_{\text{test}} = h_{\text{test}} \circ \mathcal{D}_{\text{test}}$. This objective is now a tractable saddle point optimization, which can be solved with standard stochastic optimization techniques. The optimization alternates between optimizing the reweighting $\tau$ and the visual encoder $h_{\text{test}}$, and the dual variables $f, \lambda$. Throughout adaptation, we freeze all other parts of the recurrent state space model and only optimize the encoder. We provide more intuition for the support constraint in Appendix E.

**Calibration.** We note that naively reweighting by $\tau(\cdot)$ can cause degenerate encodings that collapse into one point. To prevent this, we regularize the support constraint by also ensuring that some set of paired "calibration" states across training and testing domains share the same encoding. We collect paired trajectories in the training and testing domains using actions generated by an exploration policy, and minimize the $\ell_2$ loss between the training and testing encoding of each pair of observations. We defer the details of the complete optimization to Appendix C.

## 5 Experimental Evaluation

We conduct empirical experiments to answer the following research questions: (1) Does RePo enable learning in dynamic, distracted environments with spurious variations? (2) Do representations learned by RePo quickly adapt to new environments with test time adaptation? (3) Does RePo help learning in static, but diverse and cluttered environments?

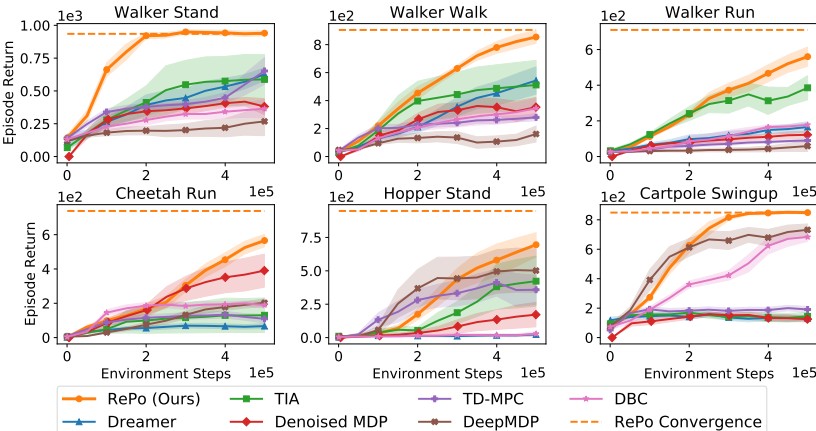

Figure 5: Results on distracted DeepMind control environments. These environments have spurious variations, and `RePo` is able to successfully learn in all of them, both faster and achieving higher asymptotic returns than prior representation learning methods.

**Evaluation domains** We evaluate our method primarily in three different settings. (1) **Distracted DeepMind Control Suite** [64, 63] is a variant of DeepMind Control Suite where the static background is replaced with natural videos (Fig. 4). For adaptation experiments, we train agents on static undistracted backgrounds and adapt them to distracted variants. (2) **Realistic Maniskill** is a benchmark we constructed based on the Maniskill2 benchmark [15], but with realistic backgrounds from [3] to simulate learning in a diverse range of human homes. We solve three tasks - LiftCube, PushCube, and TurnFaucet in a variety of background settings. (3) **Lazy TurtleBot** is a real-world robotic setup where a TurtleBot has to reach some goal location from egocentric observations in a furnished room. However, there are two TVs playing random videos to distract the "lazy" robot. We provide more details about evaluation domains in Appendix D.

**Baselines** We compare our method with a number of techniques that explicitly learn representations and use them for learning control policies. (1) **Dreamer** [17] is a state-of-the-art visual model-based RL method that learns a latent representation by reconstructing images. (2) **TIA** [9] renders Dreamer more robust to visual distractors by using a separate dynamics model to capture the task-irrelevant components in the environment. (3) **Denoised MDP** [56] further learns a factorized latent dynamics model that disentangles controllability and reward relevance. (4) **TD-MPC** [22] trains a latent dynamics model to predict the value function and uses a hybrid planning method to extract a policy. (5) **DeepMDP** [12] is a model-free method that learns a representation by predicting dynamics and reward, and then performs actor-critic policy learning on the learned representation. (6) Deep Bisimulation for Control **DBC** [63] is model-free algorithm which encodes images into a latent space that preserves the bisimulation metric.

We also compare with a number of techniques for test-time adaptation of these representations. (1) **calibrated distribution matching**, a variant of the method proposed in Section 4.1, using a distribution matching constraint rather than a support matching one, (2) **uncalibrated support matching**, a variant of the method proposed in Section 4.1, using a support matching constraint but without using paired examples, (3) **uncalibrated distribution matching**, a variant of the method proposed in Section 4.1, using a distribution matching constraint, but without using paired examples, (4) invariance through latent alignment **ILA** [61], a technique for test-time adaptation of representations with distribution matching and enforcing consistency in latent dynamics, (5) **calibration**, a baseline that only matches the encodings of paired examples.

**Does `RePo` learn behaviors in environments with spurious variations?** We evaluate our method's ability to ignore spurious variations on a suite of simulated benchmark environments with dynamic visual backgrounds (Fig. 4); these are challenging because uncontrollable elements of the environment visually dominate a significant portion of the scene. Fig. 5 shows our method outperforms the baselines across six Distracted DeepMind Control environments, both in terms of learning speed and asymptotic performance. This implies that our method successfully learns latent representations resilient to spurious variations. Dreamer [17] attempts to reconstruct the dynamic visual distractors which is challenging in these domains. TIA [9] and Denoised MDP [56] see occasional success when

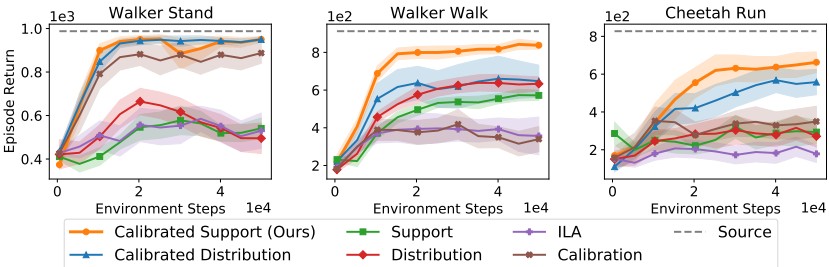

Figure 6: Results on adaptation from static environments to dynamic environments in Deepmind control. `RePo` with calibrated support constraints outperforms ablations and previous techniques for domain adaptation.

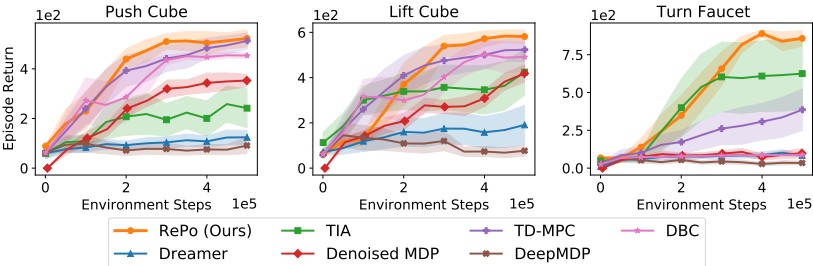

Figure 7: Results of training agents on varying static environments in Maniskill [15]. `RePo` is able to learn more quickly and efficiently than alternatives even in static domains.

they dissociate the task-relevant and irrelevant components, but they suffer from high variance and optimization failures. TD-MPC [22] is affected by spurious variations as its representations are not minimal. The model-free baselines DeepMDP [12] and DBC [63] exhibit lower sample efficiency on the more complex domains despite performing well on simpler ones.

To further validate `RePo`'s ability to handle spurious variations in the real world, we evaluate its performance on Lazy TurtleBot, where a mobile robot has to navigate around a furnished room to reach the goal from egocentric observations (Fig. 4). To introduce spurious variations, we place two TVs playing random Youtube videos along the critical paths to the goal. As shown in Table. 1, `RePo` is able to reach the goal with nontrivial success within 15K environment steps, whereas Dreamer fails to reach the goal. We provide details about the setup in Appendix. D.

Table 1: Results on Lazy TurtbleBot at 15K environment steps. `RePo` achieves nontrivial success whereas Dreamer fails to reach the goal.

|  | Success | Return |
|---|---|---|
| `RePo` (Ours) | **62.5%** | **-24.3** |
| Dreamer [17] | 0.0% | -61.7 |

**Do representations learned by `RePo` transfer under distribution shift?** We evaluate the effectiveness of the test-time adaptation method described in Section 4.1 on three DeepMind Control domains: Walker Stand, Walker Walk, and Cheetah Run. We train the representation in environments with *static backgrounds*, and adapt the representation to domains with *natural video distractors* (as shown in Fig. 4). For methods that use calibration between the source and target environments, we collect 10 trajectories of paired observations. Results are shown in Fig. 6. `RePo` shows the ability to adapt quickly across all three domains, nearly recovering the full training performance within 50k steps. Performance degrades if we replace the support constraint with a distribution matching objective, as it is infeasible to match distributions with the test-time distribution having insufficient exploration. We also observe that by removing the calibration examples, both support constraint and distribution perform worse as the distributions tend to collapse. We found the addition of dynamics consistency in ILA to be ineffective. Nor is calibration alone sufficient for adaptation.

**Does `RePo` learn across diverse environments with varying visual features?** While the previous two sections studied learning and adaptation in dynamic environments with uncontrollable elements, we also evaluate `RePo` on it's ability to learn in a *diverse* range of environments, each with a realistic and cluttered static background. Being able to learn more effectively in these domains suggests that `RePo` focuses it's representation capacity on the important elements of the task across environments, rather than trying to reconstruct the entire background for every environment.

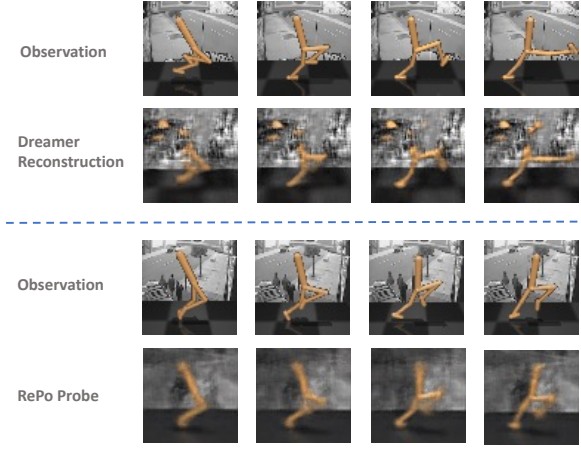

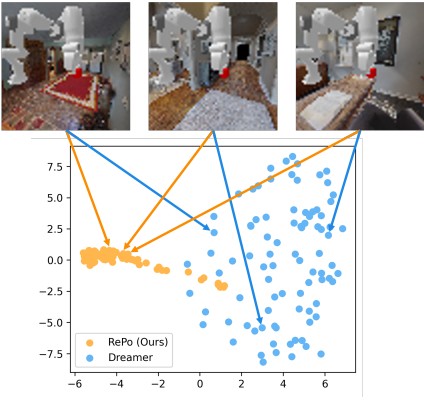

Figure 9: Probing representations learned by `RePo` shows that the background is largely ignored, while [17] tries to reconstruct it at the cost of the agent prediction.

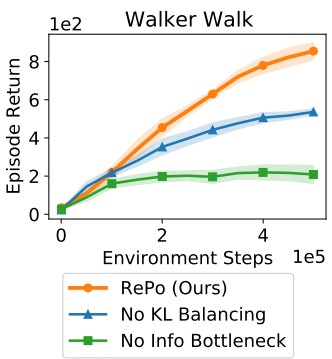

Figure 10: Top two principle components of `RePo` and Dreamer's latent representations across different backgrounds. `RePo`'s latent representation is more compact than Dreamer's, which enables data sharing.

We test on three robotic manipulation tasks - LiftCube, PushCube, and TurnFaucet with realistic backgrounds depicted in Fig. 4. As shown in Fig. 7, our method achieves saturating performance across all three tasks. Dreamer [17] spends its representation capacity memorizing backgrounds and is unable to reach optimal task performance. TIA [9] suffers from high variance and occasionally fails to dissociate task-relevant from task-irrelevant features. Denoised MDP [56], TD-MPC [22], and DBC [63] learn to ignore the background in two of the tasks but generally lag behind `RePo` in terms of sample efficiency. DeepMDP [12] fails to learn meaningful behavior in any task.

**Visualizing representations learned by** `RePo`    To decipher our representation learning objective, we probe the learned representations by post-hoc training a separate image decoder to reconstruct image observations from the latents. We visualize the results in Fig. 9 and compare them with Dreamer reconstructions [17]. Our representation contains little information about background but is capable of reconstructing the agent, implying that it contains only task-relevant information.

In addition to probing, we qualitatively compare the latent states of `RePo` and Dreamer by visualizing their top two principal components. We collect the same trajectory across all backgrounds in Maniskill and visualize the final recurrent latent state inferred by `RePo` and Dreamer respectively. As shown in Fig. 10, `RePo` produces more compact latent representations than Dreamer, meaning the latent states encode less information about background variations. This enables `RePo` to share data across different backgrounds, explaining its superior sample efficiency compared to baselines.

Figure 8: Ablating objectives showing the importance of information bottleneck and KL balancing described in Section 4.

**Ablation experiments**    We conduct ablation experiments to determine the effect of hyperparameters in Fig. 8. As we can see, the performance is crucially dependent on the information bottleneck $\epsilon$, as well as KL balancing. We refer readers to Appendix E for a more thorough discussion.

## 6  Discussion

This work presents `RePo`, a technique for learning parsimonious representations that are resilient to spurious variations. Our representation is effective on learning in dynamic, distracted environments. And while the representation is subject to degradation under distribution shift, it can be quickly adapted to new domains by a semi-supervised test-time adaptation procedure. A limitation of our method is that the learned dynamics model is no longer task-agnostic, as it only captures task-relevant information. This can be potentially addressed by simultaneously predicting multiple reward objectives. Our framework opens up several interesting directions for future research, such as: can a multi-task variant of `RePo` allow for representations applicable to a some distribution of tasks? Can we apply our algorithm in a continual learning setup? We believe our method holds promise in these more general settings, especially for real robots deployed into dynamic, human-centric environments.

## Acknowledgments and Disclosure of Funding

We would like to thank Marius Memmel, Max Balsells, and many other members of the WEIRD Lab at University of Washington for valuable feedback and discussions.

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

# Supplementary Materials for
# "RePo: Resilient Model-Based Reinforcement Learning by Regularizing Posterior Predictability"

## A    Algorithm Pseudocode

---

**Algorithm 1** Resilient Model-Based RL by Regularizing Posterior Predictability (RePo)

---

1: Initialize dataset $\mathcal{D}_{\text{buf}}$ with $S$ random seed episodes.
2: Initialize neural network parameters $\theta, \phi, \psi$ randomly and set dual variable $\beta = \beta_0$.
3: **while** not converged **do**
4:    **for** update step $c = 1 \ldots C$ **do**
5:       // Dynamics Learning
6:       Draw $B$ data sequences $\{(o_t, a_t, r_t)\}_{t=1}^{T} \sim \mathcal{D}_{\text{buf}}$.
7:       Encode images $x_{1:T} = h_\theta(o_{1:T})$.
8:       Infer prior and posterior distributions $q_\theta(z_t \mid z_{t-1}, a_{t-1}), p_\theta(z_t \mid z_{t-1}, a_{t-1}, x_t)$.
9:       Sample latent states $z_t \sim p_\theta(z_t \mid z_{t-1}, a_{t-1}, x_t)$ and infer reward distributions $q_\theta(r_t \mid z_t)$.
10:      Compute the Lagrangian objective in (4.4)
11:      Update $\theta$ and $\beta$ using dual gradient descent.
12:      // Behavior Learning
13:      Imagine trajectories $\{(z_\tau, a_\tau)\}_{\tau=t}^{t+H}$ from each $z_t$.
14:      Predict rewards $r_\tau \sim q_\theta(r_\tau \mid z_\tau)$, values $v_\tau \sim V_\psi(v_\tau \mid z_\tau)$, actions $a_\tau \sim \pi_\phi(a_\tau \mid z_\tau)$.
15:      Update $\phi$ and $\psi$ using actor-critic learning.
16:    // Data Collection
17:    **for** time step $t = 1 \ldots N$ **do**
18:      Compute $z_t \sim p_\theta(z_t, z_{t-1}, a_{t-1}, h_\theta(o_t))$ from history and current observation.
19:      Compute $a_t \sim \pi_\phi(a_t \mid z_t)$ from policy.
20:      Execute $a_t$ in environment and collect $o_{t+1}, a_{t+1}$.
21:    Add experience to dataset $\mathcal{D}_{\text{buf}} \leftarrow \mathcal{D}_{\text{buf}} \cup \{o_t, a_t, r_t\}_{t=1}^{N}$.

---

**Algorithm 2** Semi-Supervised Adaptation of Visual Encoder Using Support Constraint

---

1: Fix training encoder $h_\theta$, model $p_\theta$, policy $\pi_\phi$ and replay buffer $\mathcal{D}_{\text{train}}$.
2: Initialize test-time replay buffer $\mathcal{D}_{\text{test}}$ with $S$ random seed episodes.
3: Initialize test-time encoder $h_{\theta'}$ by setting $\theta' = \theta$.
4: Initialize neural networks $\tau_\rho$ and $f_\omega$ randomly and set dual variable $\lambda = \lambda_0$.
5: Collect $N$ calibration trajectories with paired observations $\mathcal{D}_{\text{cal}} \leftarrow \{(o_t^{\text{train}}, o_t^{\text{test}}, a_t, r_t)\}_{t=1}^{T}$ using an exploration policy.
6: **while** not converged **do**
7:    **for** update step $c = 1 \ldots C$ **do**
8:      // Adaptation
9:      Draw $B$ observations from training and test buffers $o_{1:B}^{\text{train}} \sim \mathcal{D}_{\text{train}}, o_{1:B}^{\text{test}} \sim \mathcal{D}_{\text{test}}$.
10:      Encode images using respective encoders $x_{1:B}^{\text{train}} = h_\theta(o_{1:B}^{\text{train}}), x_{1:B}^{\text{test}} = h_{\theta'}(o_{1:B}^{\text{test}})$.
11:      Compute the Lagrangian objective in (4.7)
12:      Draw $B$ pairs of observations from the calibration buffer $\{(o_i^{\text{cal\_train}}, o_i^{\text{cal\_test}})\}_{i=1}^{B} \sim \mathcal{D}_{\text{cal}}$.
13:      Encode images using encoders $x_{1:B}^{\text{cal\_train}} = h_\theta(o_{1:B}^{\text{cal\_train}}), x_{1:B}^{\text{cal\_test}} = h_{\theta'}(o_{1:B}^{\text{cal\_test}})$.
14:      Compute MSE loss between calibration encodings and add to the Lagrangian objective.
15:      Update $\theta', \rho, \omega$ and $\lambda$ using dual gradient descent.
16:    // Data Collection
17:    **for** time step $t = 1 \ldots N$ **do**
18:      Compute $z_t \sim p_\theta(z_t, z_{t-1}, a_{t-1}, h_{\theta'}(o_t))$ from history and current observation.
19:      Compute $a_t \sim \pi_\phi(a_t \mid z_t)$ from policy.
20:      Execute $a_t$ in environment and collect $o_{t+1}, a_{t+1}$.
21:    Add experience to dataset $\mathcal{D}_{\text{test}} \leftarrow \mathcal{D}_{\text{test}} \cup \{o_t, a_t, r_t\}_{t=1}^{N}$.

---

# B Derivations

## B.1 `RePo` **objective**

Recall that $\mathcal{D}_{\text{buf}}$ denote the distribution over experienced actions, observations and rewards from the environment $((a_{1:T}, o_{1:T}, r_{1:T}) \sim \mathcal{D}_{\text{buf}})$. For $p \in \mathscr{P}_{\text{post}}$ (our class of posteriors), let $\mathbb{E}_{p,h}$ denote expectation of $(a_{1:T}, o_{1:T}, r_{1:T}) \sim \mathcal{D}_{\text{buf}}$, $x_t = h(o_t)$ and the latents $z_{t+1} \sim p(\cdot \mid z_t, a_t, x_{t+1})$ drawn from the latent posterior, with the initial latent $z_0 \sim p_0$.

We derive a variational lower bound for the following information bottleneck objective from eq. (4.1).

$$\max_{p,h} \mathrm{I}_{p,h}(z_{1:T}; r_{1:T} \mid a_{1:T}) \ \text{ s.t. } \ \mathrm{I}_{p,h}(z_{1:T}; o_{1:T} \mid a_{1:T}) < \epsilon$$

where the mutual information is with respect to the distribution described by the expectation $\mathbb{E}_{p,h}$ above. The goal is to learn a latent representation that maximally predicts the dynamics and reward while sharing minimal information with the observations. Notice that, as defined, $p$ is only a latent dynamics model. Thus, for the derivation we will introduce $\tilde{p}$ as corresponding to the distribution of *all variables* under $\mathbb{E}_{p,h}$, which in particular takes into account the distribution from $\mathcal{D}_{\text{buf}}$.

**Definition B.1.** We let $\tilde{p}$ denote the joint distribution of $(z_{1:T}, a_{1:T}, x_{1:T}, o_{1:T})$ under $\mathbb{E}_{p,h}$.

**Lower bound on reward prediction.** We first derive a variational lower bound for the objective $\mathrm{I}_{p,h}(z_{1:T}; r_{1:T} \mid a_{1:T})$. For our derivation, we make a simplifying assumption:

**Assumption B.1.** Assume that there is a function $p_{\text{r}}(r_t \mid z_t)$ such that $\tilde{p}(r_t \mid z_{1:t}, a_{1:t}) = p_{\text{r}}(r_t \mid z_t)$.

The simplifying assumption enforces that $z_t$ is sufficient for the reward given past latents, actions, and the current action $a_t$. We remark that Assumption B.1 is not strictly necessary, and the following lower bound we derive is still valid if it fails; it may just be very loose. The derivation begins as follows:

$$
\begin{aligned}
& \mathrm{I}_{p,h}(z_{1:T}; r_{1:T} \mid a_{1:T}) \\
= \ & \mathbb{E}_{p,h}[\log \tilde{p}(r_{1:T} \mid z_{1:T}, a_{1:T})] + \mathrm{H}_{p,h}(r_{1:T} \mid a_{1:T}) \\
\overset{\pm}{=} \ & \mathbb{E}_{p,h}[\log \tilde{p}(r_{1:T} \mid z_{1:T}, a_{1:T})] \\
= \ & \mathbb{E}_{p,h}\left[\sum_{t=1}^{T} \log p_{\text{r}}(r_t \mid z_t, a_t))\right] && \text{(Assumption B.1)} \\
= \ & \mathbb{E}_{p,h}\left[\sum_{t=1}^{T} \log q_{\text{r}}(r_t \mid z_t))\right] + \sum_{t=1}^{T} \mathrm{D}_{\text{KL}}(p_{\text{r}}(r_t \mid z_t) \parallel q_{\text{r}}(r_t \mid z_t)) \\
\geq \ & \mathbb{E}_{p,h}\left[\sum_{t=1}^{T} \log q_{\text{r}}(r_t \mid z_t))\right],
\end{aligned}
$$

where above $\mathrm{H}_{p,h}$ denotes an entropy term under distribution $\mathbb{E}_{p,h}$, and $\overset{\pm}{=}$ denotes equality up to a constant that does not depend on our choice of parametrization $p$, which only dictates latents $z_t$. In the fourth line, we add and subtract $\log q_{\text{r}}(r_t \mid z_t)$ in each term of the summation. The last step uses the nonnegativity of KL divergence.

**Upper bound on dynamic compression.** We proceed to derive an upper bound for the constraint $\mathrm{I}_{p,h}(z_{1:T}; o_{1:T} \mid a_{1:T})$. We make a a simplifying assumption analogous to Assumption B.1; again, this assumption can also discarded, but illustrates that our lower bound is sharper when this assumption holds.

**Assumption B.2.** We assume that there is a function $p_{\text{z}}(z_{t+1} \mid z_t, a_t)$ such that $\tilde{p}(z_{t+1} \mid z_{1:t}, a_{1:t}) = p_{\text{z}}(z_{t+1} \mid z_t, a_t)$.

With Assumption B.2, we then invoke a variational upper bound which replaces $p_z(z_{t+1} \mid z_t, a_t)$ with a variational approximation $q_z(z_{t+1} \mid z_t, a_t)$:

$$
\begin{aligned}
&\mathrm{I}_{p,h}(z_{1:T}; o_{1:T} \mid a_{1:T}) \\
=\ & \mathbb{E}_{p,h}[\log \tilde{p}(z_{1:T} \mid o_{1:T}, a_{1:T}) - \log \tilde{p}(z_{1:T} \mid a_{1:T})] \\
=\ & \mathbb{E}_{p,h}\left[\left(\sum_{t=1}^{T} \log p(z_{t+1} \mid z_t, a_t, x_{t+1})\right) - \log \tilde{p}(z_{1:T} \mid a_{1:T})\right] \quad \text{(definition of latent dynamics)} \\
=\ & \mathbb{E}_{p,h}\left[\sum_{t=1}^{T} \log p(z_{t+1} \mid z_t, a_t, x_{t+1}) - \sum_{t=1}^{T} \log p_z(z_{t+1} \mid z_t, a_t)\right] \quad \text{(Assumption B.2)} \\
=\ & \mathbb{E}_{p,h}\left[\sum_{t=1}^{T} \log p(z_{t+1} \mid z_t, a_t, x_{t+1}) - \sum_{t=1}^{T} \log q_z(z_{t+1} \mid z_t, a_t)\right] - \sum_{t=1}^{T} \mathrm{D}_{\mathrm{KL}}(p_z(z_{t+1} \mid z_t, a_t) \,\|\, q_z(z_{t+1} \mid z_t, a_t)) \\
\leq\ & \mathbb{E}_{p,h}\left[\sum_{t=1}^{T} \log p(z_{t+1} \mid z_t, a_t, x_{t+1}) - \sum_{t=1}^{T} \log q_z(z_{t+1} \mid z_t, a_t)\right],
\end{aligned}
$$

where in the last two lines we add and subtract $q_z(z_{t+1} \mid z_t, a_t)$ and apply the nonnegativity of KL divergence. In the third line, we use the fact that our transition dynamics model is Markov, with

$$
\tilde{p}(z_{t+1} \mid o_{1:t}, a_{1:t}, z_{1:t}) = p(z_{t+1} \mid h(o_{t+1}), a_t, z_t) = p(z_{t+1} \mid x_{t+1}, a_t, z_t), \tag{B.1}
$$

where under $\mathbb{E}_{p,h}$, $x_t \equiv h(o_t)$ holds by definition. Notice that the same identity would be true if instead we tried to lower bound the mutual information $\mathrm{I}_{p,h}(z_{1:T}; o_{1:T} \mid a_{1:T})$, showing that our objective is agnostic to whether we consider compressing information with respect to $x_{1:T}$ or $o_{1:T}$. This may be seem suprising, but is a simple consequence of choosing a latent dynamics model which depends on $o_{1:T}$ only through $x_{1:T}$.

**Combining the derivations.** Combining these results and writing in Lagrangian form, we arrive at the objective in eq. (4.4)

$$
\max_{p, q_r, q_z, h} \min_{\beta} \mathbb{E}_{p,h}\left[\sum_{t=1}^{T} \log q_r(r_t \mid z_t)\right] + \beta\left(\mathbb{E}_{p,h}\left[\sum_{t=0}^{T-1} \mathrm{D}_{\mathrm{KL}}(p(\cdot \mid z_t, a_t, x_{t+1}) \,\|\, q_z(\cdot \mid z_t, a_t))\right] - \epsilon\right).
$$

## B.2 Tractable objective for support constraint

We derive a tractable variant for the support matching objective in eq. (4.6):

$$
\min_{\tau(\cdot) \geq 0, h_{\text{test}}(\cdot)} \mathrm{D}(\tau \cdot \mathcal{P}_{\text{train}} \,\|\, \mathcal{P}_{\text{test}}) \quad \text{s.t.} \quad \mathbb{E}_{x \sim \mathcal{P}_{\text{train}}}[\tau(x)] = 1,
$$

where $\mathcal{P}_{\text{train}}$ denotes the distribution of image encodings at training time and $\mathcal{P}_{\text{test}}$ denotes the distribution of image encodings at test time. $\tau$ is a reweighting function and the constraint $\mathbb{E}_{x \sim \mathcal{P}_{\text{train}}}[\tau(x)] = 1$ ensures that $\tau \cdot \mathcal{P}_{\text{train}}$ is a valid distribution. Let $\mathrm{D}_\phi$ be any $f$-divergence, we have

$$
\begin{aligned}
&\min_{\tau(\cdot) \geq 0, h_{\text{test}}(\cdot)} \mathrm{D}_\phi(\tau \cdot \mathcal{P}_{\text{train}} \,\|\, \mathcal{P}_{\text{test}}) \\
=\ & \min_{\tau(\cdot) \geq 0, h_{\text{test}}(\cdot)} \int_x p_{\text{test}}(x) \phi\left(\frac{p_{\text{train}}(x) \cdot \tau(x)}{p_{\text{test}}(x)}\right) \quad \text{(definition of $f$-divergence)} \\
=\ & \min_{\tau(\cdot) \geq 0, h_{\text{test}}(\cdot)} \int_x p_{\text{test}}(x) \left\{\max_{f(\cdot)} \left[\frac{p_{\text{train}}(x) \cdot \tau(x)}{p_{\text{test}}(x)} f(x) - \phi^*(f(x))\right]\right\} \quad \text{(by convex conjugacy)} \\
=\ & \min_{\tau(\cdot) \geq 0, h_{\text{test}}(\cdot)} \max_{f(\cdot)} \int_x p_{\text{test}}(x) \left[\frac{p_{\text{train}}(x) \cdot \tau(x)}{p_{\text{test}}(x)} f(x) - \phi^*(f(x))\right] \\
& \hspace{8cm} \text{(by interchangeability principle [4])} \\
=\ & \min_{\tau(\cdot) \geq 0, h_{\text{test}}(\cdot)} \max_{f(\cdot)} \mathbb{E}_{\mathcal{P}_{\text{train}}}[\tau(x) \cdot f(x)] - \mathbb{E}_{\mathcal{P}_{\text{test}}}[\phi^*(f(x))]
\end{aligned}
$$

We follow [4] and use the $\chi^2$ divergence where $\phi(x) = (x-1)^2$ and $\phi^*(y) = y + \frac{y^2}{4}$. This gives

$$
\min_{\tau(\cdot) \geq 0, h_{\text{test}}(\cdot)} \max_{f(\cdot)} \mathbb{E}_{\mathcal{P}_{\text{train}}}[\tau(x) \cdot f(x)] - \mathbb{E}_{\mathcal{P}_{\text{test}}}\left[f(x) + \frac{1}{4} f(x)^2\right].
$$

Incorporating the constraint as a Lagrange multiplier, we arrive at the objective in eq. (4.7).

$$\min_{\tau(\cdot) \, \geq \, 0, h_{\text{test}}(\cdot)} \max_{f(\cdot), \lambda} \mathbb{E}_{\mathcal{P}_{\text{train}}}[\tau(x) \cdot f(x)] - \mathbb{E}_{\mathcal{P}_{\text{test}}}\left[f(x) + \frac{1}{4}f(x)^2\right] + \lambda(\mathbb{E}_{\mathcal{P}_{\text{train}}}[\tau(x)] - 1).$$

## C  Implementation Details

**Model architecture**   We base our implementation of RePo on Dreamer [17]. We fix the image size to $64 \times 64$ and parameterize the image encoder using a 4-layer CNN with $\{32, 64, 128, 256\}$ channels, kernel size 4, stride 2, and ReLU activation. This results in an embedding size of 1024. The recurrent state space model is parametrized by a GRU operating on deterministic beliefs. Given the current belief, state (sampled from either prior or posterior), and action, the GRU recurrently predicts the next belief. We parameterize the prior $q_z(z_{t+1} \mid z_t, a_t)$ as a Gaussian whose mean and standard deviation are predicted by passing the next belief through a 2-layer MLP. Similarly, we parametrize the posterior $p(z_{t+1} \mid z_t, a_t, x_{t+1})$ as a Gaussian whose mean and standard deviation are predicted by passing the next belief along with the next image embedding through a 2-layer MLP. We set the belief size to 200 and the state size to 30. As in Dreamer, we use both deterministic belief and stochastic state as inputs to the prediction heads, including the reward model, policy, and value function. The reward model and value function are 4-layer MLPs. The policy is a 4-layer MLP outputting a squashed Gaussian distribution. We use 200 hidden units per layer and ELU activation for all MLPs.

Table 2: Hyperparameters for evaluation tasks

|  | $\beta_0$ | $\epsilon$ | $r$ |
|---|---|---|---|
| Walker | 1e-5 | 3 | 5 |
| Cheetah | 1e-5 | 3 | 5 |
| Hopper | 1e-4 | 1 | 3 |
| Cartpole | 1e-4 | 3 | 4 |
| Maniskill | 1e-4 | 3 | 4 |
| TurtleBot | 1e-5 | 3 | 5 |

**Optimization**   We train the RL agent in an online setting, performing 100 training steps for every 500 environment steps (except for TurtleBot which trains every 100 environment steps). In each training step, we sample 50 trajectories of length 50 from the replay buffer and optimize the RePo objective using the prior and posterior inferred from these trajectories. We then perform behavior learning by using the posterior states as initial latent states and rolling out the policy for 15 steps in the dynamics model. To balance bias and variance, we train the value function to predict the generalized value estimate [17] with $\lambda = 0.95$ and discount factor $\gamma = 0.99$. We optimize all components with the Adam [29] optimizer. The image encoder, recurrent state space model, and reward model share the same learning rate of 3e-4. The policy and value function use a learning rate of 8e-5. Specific to our method, we initialize the Lagrange multiplier as $\beta = \beta_0$ and set its learning rate to 1e-4. We cast the KL balancing parameter $\alpha$ to the ratio $r$ between the number of prior training steps to posterior training steps, where $r$ translates to $\alpha = \frac{r}{r+1}$. We tune the initial Lagrange multiplier $\beta_0$, target KL $\epsilon$, and KL balancing ratio $r$ on our evaluation tasks and report the best hyperparameters in Table 2.

**Baselines**   We implement Dreamer [17] and TIA [9] on top of the RePo codebase and tune their hyperparameters on the evaluation tasks. For DBC and DeepMDP, we use the official implementation along with their reported hyperparameters. We use a determinisitic transition model for DBC as we find it to perform better than using a stochastic transition model.

**Adaptation**   We parametrize the reweighting function $\tau$ as 2-layer MLP with 256 hidden units in each layer and ReLU activation. The dual function $f$ is analogous to the discriminator in a GAN architecture [14]. To prevent vanishing gradient, we parameterize $f$ using a variational discriminator bottleneck [40] with 2 hidden layers of size 256 and a bottleneck layer of size 64. Prior to training, we collect 10 calibration trajectories with corresponding observations from both the source and the target domains, using an expert policy as an approximation for an exploration policy. We initialize the test-time encoder with the weights of the training encoder and adapt it online, performing 100

adaptation steps for every 500 environment steps taken. In each adaptation step, we sample 2500 observations from the offline source buffer and online target buffer respectively, and optimize the tractable support constraint objective (4.7). In addition, we sample 2500 paired observations from the calibration data and minimize the $\ell_2$ loss between their encodings. All components are optimized using Adam [29] optimizer. We use a learning rate of 3e-4 for $h_{\text{test}}$, 5e-5 for $\tau$, 1e-4 for $f$, and 5e-3 for the Lagrange multiplier $\lambda$ initialized to 1e-4.

**Adaptation baselines**  We reuse the same model architecture for all variants of our proposed method. For baselines involving a distribution matching objective, we optimize the standard GAN [14] objective to minimize the Janson-Shannon divergence between training and test-time encoding distributions. To enforce dynamics consistency for the ILA [61] baseline, we forward the image encodings through the RSSM to get the latent states and compute dynamics violation as the KL divergence between the prior and the posterior. We minimize dynamics violation by backpropagating the gradient through the model to the encodings.

# D   Environment Details

**Distracted DeepMind Control**  To evaluate the ability of RePo to learn in dynamic environments, we use the distracted DeepMind Control Suite proposed in [63]. Specifically, we replace the static background in standard DeepMind Control environments [51] with grayscale videos from the Kinectics-400 Dataset [28]. We use a time limit of 1000 and an action repeat of 2 for all environments. We evaluate all methods with 4 random seeds.

**Realistic Maniskill**  We evaluate the ability of RePo to learn across diverse scenarios on 3 manipulation tasks adapted from the ManiSkill 2 benchmark [15]:

- **Push Cube**: the goal is to push a cube to reach a position on the floor.
- **Lift Cube**: the goal is to lift a cube above a certain height.
- **Turn Faucet**: the goal is to turn the faucet to a certain angle.

To simulate real-world scenarios, we replace the default background with realistic scenes from the Habitat Matterport dataset [43]. We curate 90 different scenes and randomly load a new scene at the beginning of each episode. We use a time limit of 100 and an action repeat of 1. We evaluate all methods with 4 random seeds.

**Lazy TurtleBot**  To evaluate RePo's resilience against spurious variations in the real world, we furnish a room to mimic a typical household setup and train a TurtleBot to reach a certain goal location from egocentric observations (Fig. 4). We introduce spurious variations by placing two TVs playing random YouTube videos along the critical paths to the goal. The robot features a discrete action space with 4 actions: move forward, rotate left, rotate right, and no-op. To minimize algorithmic change, we convert the action space to a continuous 2D action space where the discrete actions correspond to $(1, 0), (0, 1), (-1, 0), (0, -1)$ respectively, and continuous actions are mapped back to the closest discrete actions in L2 distance. The reward is the negative distance to goal, where the robot's state is estimated using an overhead camera mounted on the ceiling. We apply a time limit of 100 and an action repeat of 1. At the start of training, we prefill the replay buffer with 5K offline transitions collected by sampling and reaching random goals using a state-based controller. This is followed by 10K steps of pure offline training. We then train each method online for 10K environment steps, using the state-based controller for automatic resets. Due to time constraints, we evaluate each method for 2 seeds.

# E   Additional Experiments

## E.1   Results on standard DMC environments

In Fig. 11, we provide additional comparisons to Dreamer on standard DeepMind Control tasks. We note that RePo is able to match Dreamer in asymptotic performance despite not being trained with a reconstruction objective. Our method slightly lags behind Dreamer in terms of sample efficiency, which is due to reward signals being inherently more sparse than pixel reconstruction signals.

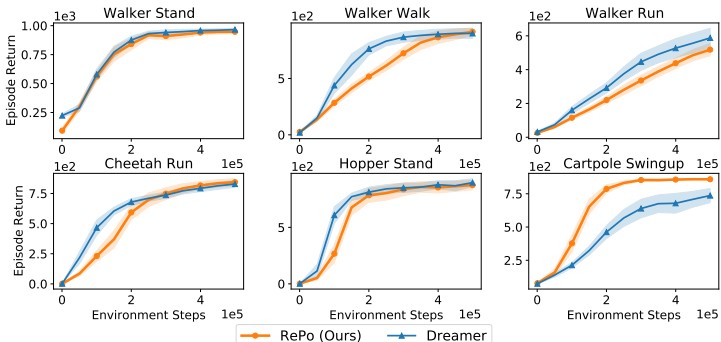

Figure 11: Comparison with Dreamer on standard DeepMind control environments. `RePo` matches Dreamer in asymptotic performance despite not being trained with a reconstruction objective.

## E.2 Results on Distracted Walker Walk with scoreboard

We evaluate `RePo` on a variant of Distracted Walker Walk which consists of a scoreboard displaying the cumulative reward of the current episode. The goal is to investigate if RePo's reward-predictive nature causes its latent representation to collapse onto predicting the score. We present the probing visualization in Fig. 12, which shows that the learned latent reconstructs the joint positions of the agent and ignores the score. This is because RePo predicts not only the current reward but also the dynamics and future rewards. The empirical performance with the scoreboard is $872.86 \pm 43.71$, while without the scoreboard it is $868.72 \pm 53.93$.

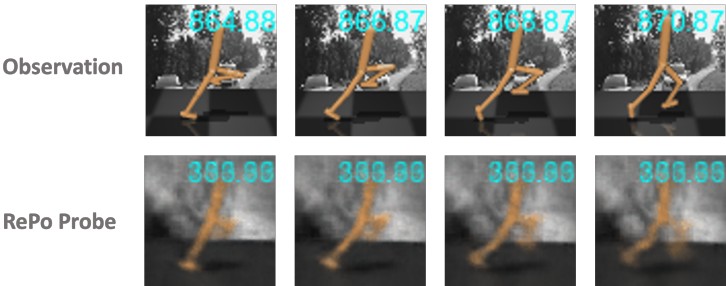

Figure 12: Visualization of `RePo`'s representation on distracted DMC with scoreboard. Since RePo predicts not only the current reward but also the dynamics and future rewards, its latent representation does not collapse to predicting the score.

## E.3 Additional ablations

We perform additional ablation experiments on distracted DMC Walker Walk to analyze the effect of hyperparameters. All other hyperparameters are fixed to those reported in Table. 2. Fig. 13 illustrates the effect of $\beta$ learning rate. When the learning rate is too high, the constraint is enforced immediately after training begins, which can lead to ineffective exploration. When the learning rate is too low, dual gradient descent converges slowly. We find the sweet spot to be a learning rate that relaxes the information bottleneck at the beginning of training and gradually tightens the bottleneck to discard redundant information in the representation. Fig. 14 show the result of ablating the KL constraint value $\epsilon$. With too small a KL target, the representation becomes degenerate and fails to capture task-relevant information, as indicated by the increase in reward loss. A large KL target, on the other hand, can result in the dynamics model being inaccurate, thereby affecting behavior learning. Finally, Fig. 15 explores the effect of KL balancing. In accordance with [19], we find that the dynamics model is generally harder to train than the representation model. Hence, a larger KL balancing parameter tends to result in better performance. However, if we train the prior too aggressively, then the posterior becomes poorly regularized and captures task-irrelevant information.

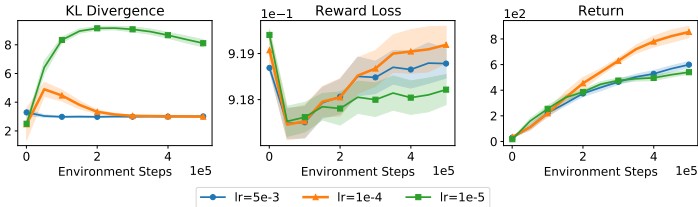

Figure 13: Ablating $\beta$ learning rate. A larger $\beta$ learning rate corresponds to faster convergence of the constraint, which can lead to ineffective exploration.

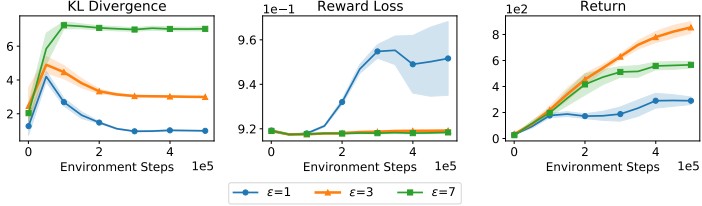

Figure 14: Ablating information bottleneck $\epsilon$. A tighter information bottleneck generally induces more parsimonious representations and more accurate dynamics models. But too tight a bottleneck can thwart reward learning.

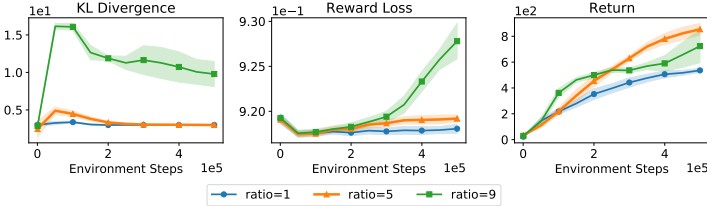

Figure 15: Ablating KL balancing ratio. The dynamics model (prior) is generally harder to train than the representation model (posterior), which suggests training the prior more frequently than the posterior, i.e. using a higher KL balancing ratio. However, training the prior too aggressively leads to poorly regularized posterior.

### E.4  An illustrative example of support constraint

We provide a didactic example to illustrate the intuition behind using a support constraint for alignment. Consider two ground truth latent distributions shown in Fig. 16. The training distribution $\mathcal{P}_{\text{train}}$ is a uniform distribution spanning $[0,6] \times [0,6]$, and the test-time distribution $\mathcal{P}_{\text{test}}$ is a uniform distribution covering exactly half the support of $\mathcal{P}_{\text{train}}$. This is to simulate insufficient online exploration at test time. We construct a nonlinear emission function $f : \mathcal{X} \rightarrow \mathcal{O} : f(x) = 0.025e^x$ to generate the observations. For illustration purposes, we use the same emission function for both training

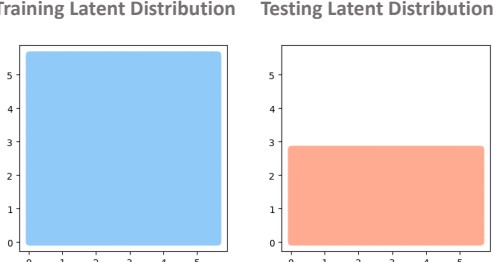

Figure 16: Ground truth latent distributions in training and testing domains.

and testing domains. Given access to a perfect training encoder, our goal is to learn a test-time encoder that recovers the ground truth encoding function (inverse of emission function) and in turn the test-time latent distribution. We compare the support constraint objective (detailed in eq. (4.7)) with a distribution matching objective minimizing the Jenson-Shannon divergence.

First, notice in Fig. 17 that naively optimizing either objective fails to recover the ground truth test-time latent distribution. The distribution matching objective "stretches" the test-time distribution to match the training distribution, whereas the support constraint only forces the former to be within the support of the latter without any additional stipulation. A common method in unsupervised domain adaptation is to enforce dynamics consistency between adjacent states, which in our toy example can be interpreted as preserving the pairwise distance between latent states. When we add in

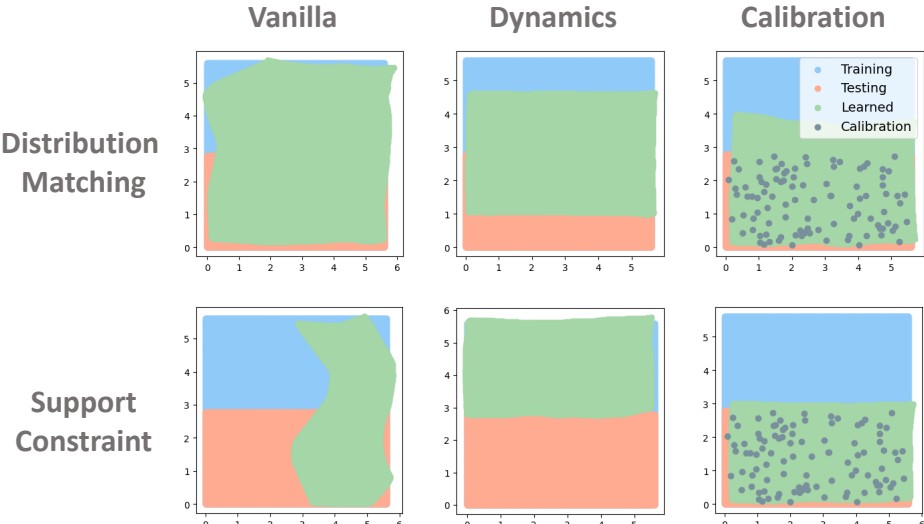

Figure 17: Comparison between distribution matching and support constraint. When there is insufficient exploration, the distribution matching object conflicts with dynamics consistency and calibration objectives, whereas support constraint conforms to these objectives.

this objective, a crucial difference between support constraint and distribution matching transpires: while the support constraint conforms to the dynamics consistency objective through the reweighting function and recovers the correct shape of the distribution, the distribution matching objective is inherently conflicting with dynamics consistency. Intuitively, the distribution matching objective attempts to stretch the distribution, whereas the dynamics consistency pulls it together. This leads to suboptimal results as shown in the middle column of the first row in Fig. 17. Still, neither method can recover the ground truth distribution by only using dynamics consistency, as it only preserves the relative positions of latents but not the global position. To address this, we propose to match the ground truth latent state for a small number of calibration examples. These calibration examples effectively serve as an anchor for the distribution. When the calibration samples have good coverage, we see the support matching objective successfully recovers the ground truth encoding function and latent distribution. Distribution matching, on the other hand, yields suboptimal solution as it still conflicts with the calibration objective.

We further investigate the case where the calibration examples do not have good coverage. In Fig. 18, we see that when the calibration examples only cover a small region of the test-time distribution, the result of optimizing the support constraint is accurate only around the calibration examples. Under this circumstance, the addition of dynamics consistency proves to be quite effective. Since (1) the distribution is anchored around the calibration samples and (2) pairwise distance is preserved, the only valid solution is recovering the ground truth latent distribution. In our DMC benchmark, we do observe degradation when we replace the exploration policy with a random policy for collecting calibration samples. Yet we find optimizing for dynamics consistency through RSSM to be of little avail. We hypothesize this is due to the recurrent architecture interfering with the optimization, and we leave it for future work to investigate potential solutions.

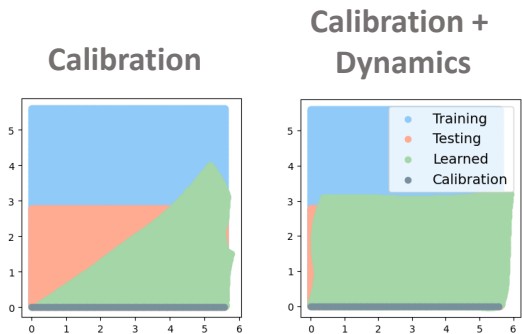

Figure 18: Support constraint with skewed calibration examples. When the calibration examples are skewed, the result of optimizing the support constraint is only accurate around the calibration examples. However, adding dynamics consistency effectively propagates the calibration signal to other states.

