# OpenReview forum: "RePo: Resilient Model-Based Reinforcement Learning by Regularizing Posterior Predictability"
_NeurIPS.cc/2023/Conference — NeurIPS 2023 spotlight_

### Official Review · Reviewer_z3JG · 2023-06-15

**Soundness:** 2 fair
**Presentation:** 2 fair
**Contribution:** 2 fair
**Rating:** 5
**Confidence:** 4

**Summary:**

This paper presents a model-based reinforcement learning (RL) method which learns a task-relevant latent-space dynamic model. It learns a latent representation containing task-relevant features by 1) maximising the mutual information between latent states and rewards whilst 2) minimising the mutual information between the latent states and observations. This contrasts approaches which reconstruct the image observations and thus learn task-irrelevant features. They highlight that their method fails to adapt to environment changes which results in a distribution shift to the observations, e.g. changes in lighting. To this end, they present a method for adapting to distribution shifts at test time. They test their method in modified versions of the DeepMind Control Suite and Maniskill with (distracted) video backgrounds.

**Strengths:**

The paper addresses an important problem which is needed for deploying model-based RL algorithms in real-world robotics scenarios.
I am not familiar with test-time adaptation for visual model-based RL, however, I do think this is an imprtant problem and the approach seems sensible.

The experiments seem to be reproducible from the implementation details in the appendix. However, I have not ran the code to check.


**Weaknesses:**

The paper's biggest weakness is its positioning against prior work.
There exist visual model-based RL methods that learn task-relevant latent representations which are not cited in this paper.
For example:
```
- Denoised MDPs
    @inproceedings{tongzhouw2022denoisedmdps,
    title={Denoised MDPs: Learning World Models Better Than The World Itself},
    author={Wang, Tongzhou and Du, Simon S. and Torralba, Antonio and Isola, Phillip and Zhang, Amy and Tian, Yuandong},
    booktitle={International Conference on Machine Learning},
    organization={PMLR},
    year={2022}
    }
- TD-MPC
    @inproceedings{Hansen2022tdmpc,
        title={Temporal Difference Learning for Model Predictive Control},
        author={Nicklas Hansen and Xiaolong Wang and Hao Su},
        booktitle={ICML},
        year={2022}
    }
- ALM
    @inproceedings{ghugare2023simplifying,
      title={Simplifying Model-based RL: Learning Representations, Latent-space Models, and Policies with One Objective},
      booktitle = {Advances in Neural Information Processing Systems},
      author={Raj Ghugare and Homanga Bharadhwaj and Benjamin Eysenbach and Sergey Levine and Ruslan Salakhutdinov},
      year={2022},
    }
```
Given the similarity of the work, I think it is important to not only discuss them in the related work but to also compare to them in the experiments section.
Especially as it is not obvious to me that this is a better approach to learning a task-relevant latent space.
I am interested to see how this method compares but I do not think this can be published without such comparisons.

The training curves should show the converged values with dashes lines. In many figures it is not possible to see where algorithms converge. For example, dreamer in Walker Stand/Walker Walk/Walker Run in Figure 5.

There are a lot of minor corrections and typos.

Minor corrections and typos:
- How many seeds were used for each experiment?
- Do the shaded regions on figures show variance or std or something else?
- Line 22 - operator should be operate
- Line 22 - the sentence doesn't read properly
  - Perhaps you could replace "and learning to operator the same arm amidst of plentiful dynamic distractors" with "and learning to operate the same arm amidst dynamic distractors".
- Eq 4.1 - $I_{p,h}(z_{1:T};x_{1:T} \mid a_{1:T})$ should be $I_{p,h}(z_{1:T}; o_{1:T} \mid a_{1:T})$
- Figures 4/7/8/9 captions should have full stops.
- Figures 1/2/3 are never referenced.
- Figure 1 is not very informative so I am unsure why it is included?
- Line 113 - brackets in expectation
- Line 123/132 - What does the $h$ subscript in $s_{h}$ represent? You introduce $s_{t}$ where the subscript is time $t$ but then it looks like you're indexing at the encoder?
- Line 84-87 - This sentence doesn't read properly.
- Line 221 - $\bar{P} : \mathcal{X} \times \mathcal{A} \rightarrow \Delta(\mathcal{X})$ should have $\times$ not $\rightarrow$?
- Line 324 - "Does representations" should be "Do representations"
- Line 360 - Incorrect use of semicolon
- Line 366 - "Can we apply our algorithm could be used in a continual learning setup?" doesn't make sense.
- Eq. 4.1/4.5/4.6 are missing full stops.
- Be consistent with Eq. vs Equation, vs Eqn. vs Eqs.
  - Line 197 - Eqn should be Eq. for consistency
  - Line 197 - Eqn should be Eq. for consistency
  - Line 177 - (4.3) should be Eq. 4.3?
  - Line 167 - (4.1) should be Eq. 4.1?
  - Line 261 - (4.5) should be Eq. 4.5?
  - Line 265 - (4.5) should be Eq. 4.5?
- Similarly, with Fig vs Fig. vs Figure.
- Parsimonious/eschew seem unnecessarily complicated words. I had to look them up... I recommend simpler words to make the paper more accessible. I asked around and 60% of people would need to look up both words.


**Questions:**

1. Are the authors aware of the literature mentioned above? Do you think you need to compare to these methods?
2. $q$ is usually used for approximate posteriors and $p$ is used for priors. Is there a reason you have swapped these around?


**Limitations:**

The authors mention that their method is not task agnostic and I think this is sufficient.

---

> ### Author Rebuttal · Authors · 2023-08-09
>
> Thank you for your careful review and constructive feedback. We address your comments below.
>
> > Given the similarity of the work, I think it is important to not only discuss them in the related work but to also compare to them in the experiments section.
>
> Thank you for suggesting these baselines. We provide additional results in Figs. 1 and 2 of the supplementary pdf showing that our method indeed outperforms Denoised MDP and TD-MPC. We hypothesize that our advantage over denoised MDP is (a) we do not get penalized for state reconstruction, (b) we do not spend representation capacity on learning a dynamics model in the task-irrelevant components of the state space, and (c) we do not need to tune multiple loss-weights as hyperparameters. TD-MPC, on the other hand, predicts the value function (but not state reconstruction). However, it has no explicit mechanism to encourage minimality of the representation. By contrast, RePo introduces a simple objective that encourages the learning of a minimal task-relevant latent representation. In addition, we propose a reward-free alignment method to handle test-time distribution shift.
>
> We omit the comparison with ALM since the authors did not evaluate their method on visual domains.
>
> > The training curves should show the converged values with dashes lines. In many figures it is not possible to see where algorithms converge.
>
> We provide the convergence returns of RePo for all Distracted DMC environments in Fig. 1. All agents converge after 1M environment steps.
>
> > How many seeds were used for each experiment?
>
> We run 4 seeds for each method on each environment.
>
> > Do the shaded regions on figures show variance or std or something else?
>
> The shaded regions denote standard error.
>
> > Figure 1 is not very informative so I am unsure why it is included?
>
> We include Fig. 1 to illustrate the motivation of our method. We will improve it in the revision by making the illustration more realistic and intuitive.
>
> > Line 123/132 - What does the h subscript in s_h represent? You introduce s_t where the subscript is time t but then it looks like you're indexing at the encoder?
>
> We made a typo on lines 123 and 132. There is no subscript on $s$. We will fix it in the revision.
>
> > q is usually used for approximate posteriors and p is used for priors. Is there a reason you have swapped these around?
>
> We follow Dreamer [1] convention and use q to denote the prior and p for the posterior.
>
> Thank you again for your thorough review! We will be sure to fix the typos in the revision.
>
> [1] Danijar Hafner, Timothy Lillicrap, Jimmy Ba, Mohammad Norouzi. Dream to Control: Learning Behaviors by Latent Imagination. ICLR 2020.

---

> > ### Comment · Reviewer_z3JG · 2023-08-13
> >
> > Thank you for taking on board our comments. I am pleased to see you have included TD-MPC and Denoised-MDP baselines as this was one of my largest concerns. I will increase my score.

---

### Official Review · Reviewer_ZoCZ · 2023-06-25

**Soundness:** 3 good
**Presentation:** 3 good
**Contribution:** 3 good
**Rating:** 5
**Confidence:** 4

**Summary:**

This paper presents an approach that aims to enhance visual model-based RL by learning state representations that filter out irrelevant information while retaining task-specific information. The proposed method achieves this by maximizing the mutual information (MI) between the current representation and future rewards, while simultaneously minimizing the MI between the representation and observations. In order to facilitate the transfer of the learned representation to new environments with distribution shifts, this paper introduces a support constraint regularization technique. This technique ensures that a specific set of "calibration" states, paired across training and testing domains, share the same encoding. The empirical results obtained demonstrate the effectiveness of the proposed method.

**Strengths:**

1. This paper addresses an important open problem in RL and is very well motivated.
2. The logic flow of the paper is good, and the presentation is clear and coherent making it a pleasant read.
3. Test-time adaptation is a reasonable and interesting topic in this field.
4. Empirical studies show the performance gain of the proposed method.

**Weaknesses:**

1. The MDP setting in this paper bears strong resemblance to the exogenous distractors setting discussed in [1, 2, 3], as well as the denoised MDP setting introduced in [4]. It is important to include a discussion of these works in the related work section. Additionally, it is worth noting that the denoised MDP approach presented in [4] also offers a model-based method for learning state representations, which should be considered as a baseline method in the experiments conducted.
2. Utilizing the recurrent state-space model (RSSM) to parameterize both p and q is a reasonable choice. However, understanding Assumption A.1 is challenging. While $z_t$ can represent past latents and past actions, it raises questions about whether $z_t$ can effectively incorporate the information from $a_t$, given that the RSSM does not explicitly model their corresponding relationship.

Typos:
1. Equation 4.1 is a bit strange. Seems the equation in Appendix A.1 is more reasonable.
2. Line 192 "parametrize" -> "parameterize"

[1]: Yonathan Efroni, Dylan J. Foster, Dipendra Misra, Akshay Krishnamurthy, John Langford: Sample-Efficient Reinforcement Learning in the Presence of Exogenous Information. COLT 2022: 5062-5127

[2]: Yonathan Efroni, Dipendra Misra, Akshay Krishnamurthy, Alekh Agarwal, John Langford: Provable RL with Exogenous Distractors via Multistep Inverse Dynamics. CoRR abs/2110.08847 (2021)

[3]: George Trimponias, Thomas G. Dietterich: Reinforcement Learning with Exogenous States and Rewards. CoRR abs/2303.12957 (2023)

[4]: Tongzhou Wang, Simon S. Du, Antonio Torralba, Phillip Isola, Amy Zhang, Yuandong Tian: Denoised MDPs: Learning World Models Better Than the World Itself. ICML 2022: 22591-22612

**Questions:**

Most important questions are presented in above.
Furthermore, what are the theoretical advantages of RePo over bisimulation-based methods? It appears that the proposed method necessitates test-time calibration to adapt to changes in distractors, whereas bisimulation-based methods can transfer their representations directly without any additional modifications. Notably, certain bisimulation approaches [1, 2] have demonstrated impressive performance not only in standard settings but also in distractor settings.

[1]: Pablo Samuel Castro, Tyler Kastner, Prakash Panangaden, Mark Rowland: MICo: Improved representations via sampling-based state similarity for Markov decision processes. NeurIPS 2021: 30113-30126

[2]: Hongyu Zang, Xin Li, Mingzhong Wang: SimSR: Simple Distance-Based State Representations for Deep Reinforcement Learning. AAAI 2022: 8997-9005

**Limitations:**

The method necessitates test-time adaptation instead of direct transfer. This requirement for additional calibration time may increase computational complexity during the evaluation stage.

---

> ### Author Rebuttal · Authors · 2023-08-09
>
> Thank you for your insightful comments and finding our paper “well-motivated” and a “pleasant read.” We address your questions below.
>
> > The MDP setting in this paper bears strong resemblance to the exogenous distractors setting discussed in [1, 2, 3], as well as the denoised MDP setting introduced in [4].
> - The work on exogenous distractors is very interesting, and we will be sure to include an extended discussion in the revision. While these works provide interesting theoretical insights, they are not deployable at the scale of our experimental environments. These works further assume that the distractor state of the MDP is entirely decoupled from the effect of control actions. This means that the exogenous MDP algorithms will learn the entire controllable part of the MDP even if it contains reward-irrelevant information.
> - The Denoised MDP refines the exogenous MDP perspective by learning a factorized latent state which disentangles the non-controllable, reward-irrelevant controllable, and reward-relevant controllable parts of the MDP. However, unlike RePo, they do not provide a mechanism to enforce minimality of all the factors. Indeed, we show that our method outperforms Denoised MDP in Figs. 1 and 2 of the supplementary pdf.
>
> > While z_t can represent past latents and past actions, it raises questions about whether z_t can effectively incorporate the information from a_t, given that the RSSM does not explicitly model their corresponding relationship.
>
> Our assumption is that the reward can be predicted from states and is independent of current action. We build this assumption into our model, which allows us to predict the dynamics, reward, value function, and policy all from the same latent state. It is straightforward to generalize our model to allow for action-dependent rewards by conditioning the reward predictor on the action. However, in our applications of interest, rewards are state dependent so we build in this information.
>
> > Furthermore, what are the theoretical advantages of RePo over bisimulation-based methods? It appears that the proposed method necessitates test-time calibration to adapt to changes in distractors, whereas bisimulation-based methods can transfer their representations directly without any additional modifications.
> - Bisimulation (Bisim) is optimal in the sense that a perfect bisimulation metric does not discard any relevant information about an MDP. However, there is no guarantee that Bisim will disregard irrelevant information. Indeed, the identity mapping induces a trivial Bisim metric without data compression. Thus, Bisim methods compress only by reducing the dimensionality of the latent variable; RePo, on the other hand, has an explicit term in its loss that encourages discarding irrelevant information by regularizing the posterior predictability of rewards/dynamics [1][2][3].
> - We note that the latent representation of our method is invariant, but the visual encoder does not generalize out of distribution, hence the need for test-time adaptation. As indicated in the invariant risk minimization [4] line of work, any approach that uses a CNN encoder to extract visual features -- Bisim included -- also suffers from the same problem and benefits from test-time adaptation.
>
> [1] Pablo Samuel Castro. Scalable methods for computing state similarity in deterministic Markov Decision Processes. AAAI 2020.
>
> [2] Amy Zhang, Rowan McAllister, Roberto Calandra, Yarin Gal, Sergey Levine. Learning invariant representations for reinforcement learning without reconstruction. ICLR 2021.
>
> [3] Amy Zhang, Clare Lyle, Shagun Sodhani, Angelos Filos, Marta Kwiatkowska, Joelle Pineau, Yarin Gal, Doina Precup. Invariant Causal Prediction for Block MDPs. ICML 2020.
>
> [4] Elan Rosenfeld, Pradeep Ravikumar, Andrej Risteski. The Risks of Invariant Risk Minimization. ICLR 2021.

---

> > ### Comment · Reviewer_ZoCZ · 2023-08-15
> > **Feedback**
> >
> > I appreciate the author's rebuttal, which includes some new empirical results. Yet, I still hope to see more comparisons between the proposed method and bisimulation approaches theoretically or empirically. I will suggest an acceptance but keep the score as-is.

---

### Official Review · Reviewer_CzT8 · 2023-07-03

**Soundness:** 4 excellent
**Presentation:** 4 excellent
**Contribution:** 4 excellent
**Rating:** 7
**Confidence:** 3

**Summary:**

The paper has two major contributions. The first one is the introduction of a model-based RL method that learns latent representations of visual RL environments that are resilient to spurious variations. The second contribution is a test-time alignment method that makes the latent representations invariant distribution shifts, thereby enabling the quick adaptation to diverse environments.

**Strengths:**

The paper proposes a novel method to address a very important problem in reinforcement learning. The RePo method is simple, elegant, and well-motivated and the discussion around eq. 4.3 makes it easy to intuitively understand it.  The presentation is something to compliment the authors for. The paper is very well written and easy to follow, despite the mathematical complexity of the underlying concepts. The relevant work section is concise, and the authors clearly relate their work with previous papers.


**Weaknesses:**

The only minor weakness that I can note is that the authors did not discuss any subtle limitations of their method, which I am sure it does have. Adding such a discussion would make the paper more credible.


**Questions:**

Should I assume that the method name RePo comes from Resilient Posterior? If so, please clarify that the first time you use it, e.g., line 40.

Are there any constraints on the environment distribution shift? On the experiments you tested on natural video distractors. What if it was something extreme like white noise?


**Limitations:**

I do not foresee any negative societal impact. Limitations were not discussed. The authors should add a small paragraph with the limitations.

---

> ### Author Rebuttal · Authors · 2023-08-09
>
> Thank you for your positive feedback and for finding our paper “simple, elegant, and well-motivated.” We address your comments below.
>
> > The only minor weakness that I can note is that the authors did not discuss any subtle limitations of their method, which I am sure it does have.
>
> We note in Section 5 that due to its reward-predicting nature, RePo’s latent representation is no longer task-agnostic, which makes it challenging to generalize to new reward functions. This could be potentially addressed by training the model to predict multiple reward objectives at once, where each objective exposes some partial view of the environment dynamics. We also note that paired calibration data used for alignment can be expensive to collect, and we hope to explore more unsupervised approaches to alignment in the future.
>
> > Are there any constraints on the environment distribution shift? On the experiments you tested on natural video distractors. What if it was something extreme like white noise?
>
> We do not put an explicit constraint on the environment distribution shift. In the case where the background is replaced by white noise, we expect both our method and baselines to perform well. This is because if the noise is random Gaussian and independent at each time step, then it contains no dynamical information and thus will not contribute as a spurious correlation. In Fig. 5 of the supplementary pdf, we compare RePo on Distracted Walker Walk with video distractor and white noise, and we do not see a difference in terms of performance.
>
> > Should I assume that the method name RePo comes from Resilient Posterior? If so, please clarify that the first time you use it, e.g., line 40.
>
> The name RePo comes from **Re**gularizing **Po**sterior Predictability. We will add a clarification in the revision.

---

### Official Review · Reviewer_zSkA · 2023-07-05

**Soundness:** 3 good
**Presentation:** 3 good
**Contribution:** 3 good
**Rating:** 7
**Confidence:** 4

**Summary:**

This work proposes a modification to Dreamer [1], wherein the authors discard the reconstructions objective, and replace it with an objective that maximizes the information contained in the latent representation about the reward, while minimizing the information about the original image. This pushes the model to learn representations that are agnostic to 'spurious variations' - changes in the input that do not affect the reward.

The authors compare the method to Dreamer [1], TIA [2] and bisimulation [3], and show that RePo deals with distracting background on DMC very well compared to other methods.

Authors also show that the method can generalize from static training data to test-time data with distractors if an additional test-time adaptation stage is introduced.

[1] Hafner D, Lillicrap T, Ba J and Norouzi M 2019 Dream to Control: Learning Behaviors by Latent Imagination _arXiv_
[2]  Fu X, Yang G, Agrawal P and Jaakkola T 2021 Learning Task Informed Abstractions
[3] Zhang A, McAllister R, Calandra R, Gal Y and Levine S 2021 Learning Invariant Representations for Reinforcement Learning without Reconstruction

**Strengths:**

- The paper is clearly written
- The proposed method is simple
- The method performs better than baselines on DMC with distractions.
- The method is able to work with distractions introduced at test-time if additional test-time adaptation stage is used

**Weaknesses:**

- This method, as authors have mentioned, is no longer task-agnostic.
- Possibly, if there's some information about the reward in the image (like a score), the model will focus on that instead

**Questions:**

- Can the authors clarify what would happen if the image itself contains information about the score.
- As a baseline, would be useful to see how Dreamer performs with distribution shfit.

**typos**
- 22 - learning to operator - > learning to operate
- 120 - to a an -> to an
- 135 - effect -> affect
- 366 - could be used should be removed

**Limitations:**

The limitations are adequately addressed in the submission.

---

> ### Author Rebuttal · Authors · 2023-08-09
>
> Thank you for your constructive comments. We address your questions below.
>
> > Can the authors clarify what would happen if the image itself contains information about the score.
>
> Since RePo not only predicts the current reward, but also the dynamics and future rewards, adding the score information would not cause it to only focus on the score. To verify this, we construct a variant of Distracted Walker Walk consisting of a scoreboard displaying the current episodic return. We train a RePo agent on this environment and probe its latent states to reconstruct the observations. As shown in Figure 3, RePo successfully reconstructs the joint positions of the agent and does not only focus on the score.
>
> > As a baseline, would be useful to see how Dreamer performs with distribution shfit.
>
> We provide the baseline comparison in Table 1. We see that Dreamer experiences a significant performance drop when the background becomes out of distribution.

---

> > ### Comment · Reviewer_zSkA · 2023-08-11
> >
> > Thank you for addressing my questions, I raise my score to 7. Good work!

---

### Author Rebuttal · Authors · 2023-08-09

We thank the reviewers for their careful reading and constructive feedback. To address the questions raised in the reviews, we conduct the following additional experiments and report the results in the supplementary pdf:
- We introduce two baselines suggested by the reviewers: Denoised MDP [1] and TD-MPC [2]. Denoised MDP learns a factorized latent dynamics model that disentangles controllability and reward relevance. However, they do not enforce minimality since they still reconstruct the state and do not optimize an information bottleneck objective. TD-MPC trains a latent dynamics model to predict the value function and uses a hybrid planning method to extract a policy. While its latent representation is task-oriented, it is not minimal and necessarily contains information about the policy. As shown in Figs. 1 and 2, RePo generally outperforms both baselines across Distracted DMC and Realistic Maniskill environments.
- We evaluate RePo on a variant of Distracted Walker Walk which consists of a scoreboard displaying the cumulative reward of the current episode. The purpose is to test if RePo’s reward-predictive nature causes its latent representation to collapse onto predicting the score. We present the probing visualization in Fig. 3, which shows that the learned latent reconstructs the joint positions of the agent and ignores the score. This is because RePo predicts not only the current reward but also the dynamics and future rewards.
- We qualitatively compare the latent states of Dreamer and RePo by visualizing their top two principal components. Specifically, we collect the same trajectory across 90 different backgrounds and visualize the final recurrent latent state inferred by Dreamer and RePo respectively. As shown in Fig. 4. RePo produces more compact latent representations than Dreamer, meaning the latent states encode less information about background variations. This enables RePo to share data across environments, which explains its superior sample efficiency compared to other baselines.
- In Fig.5, we evaluate RePo on a variant of Distracted Walker Walk with white noise as the background and observe no performance difference compared to the video distractor setting.
- As a baseline comparison, we train Dreamer agents on the original DMC environments and evaluate them on the distracted variants. From Table 1, we can see that Dreamer experiences a significant performance drop when the background becomes out of distribution.

We provide additional clarifications, explanations and discussion in the per-reviewer responses.

[1] Tongzhou Wang, Simon S. Du, Antonio Torralba, Phillip Isola, Amy Zhang, Yuandong Tian. Denoised MDPs: Learning World Models Better Than the World Itself. ICML 2022.

[2] Nicklas Hansen, Xiaolong Wang, Hao Su. Temporal Difference Learning for Model Predictive Control. ICML 2022.

---

### Decision · Program_Chairs · 2023-09-21

**Decision:**

Accept (spotlight)

**Comment:**

This paper proposes a new objective to train visual model-based RL algorithms with the aim of avoiding modelling spurious variations that occur in the image. This is a much-needed work to improve deep MBRL algorithms. The authors also propose a test-time adaptation of the encoder for handling distribution shifts. The results prove that the proposed approach is better than Dreamer and other methods.

Both reviewers with a score of 5 asked for more comparisons. The authors made some comparisons in the rebuttal pdf and also explained why some comparisons might not be necessary. I am satisfied with the answers.

I think this is an interesting research direction that is worth a spotlight.